# PBR-SR: Mesh PBR Texture Super Resolution from 2D Image Priors

**Yujin Chen**[1]  **Yinyu Nie**[1]  **Benjamin Ummenhofer**[2]  **Reiner Birkl**[2]
**Michael Paulitsch**[2]  **Matthias Nießner**[1]

[1] Technical University of Munich  [2] Intel Labs

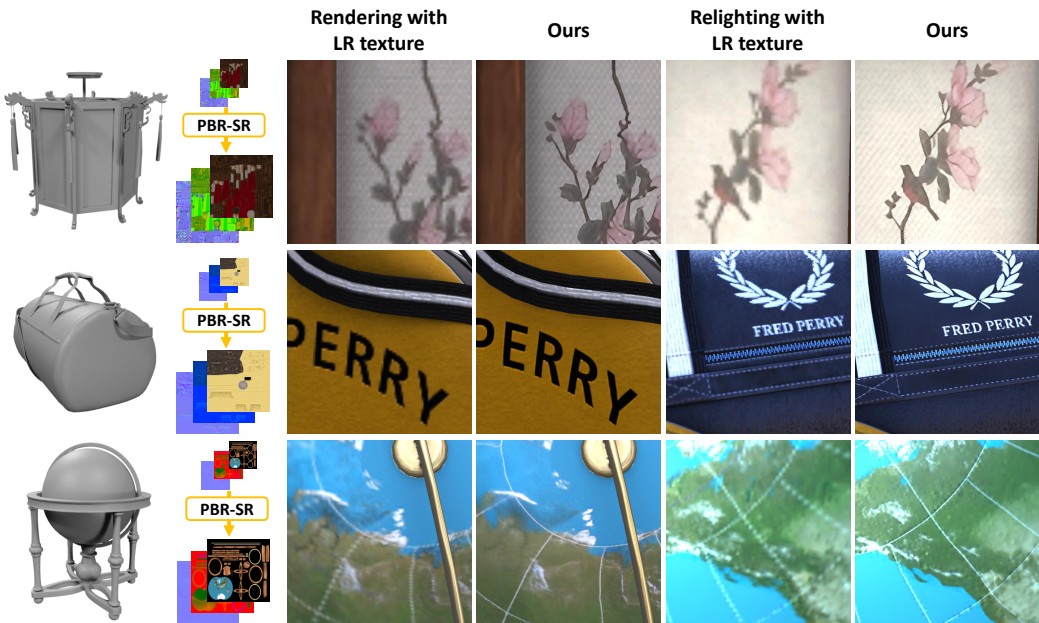

Figure 1: Given a mesh with low-resolution (LR) PBR texture maps, PBR-SR generates high-resolution (HR), high-quality PBR textures by leveraging 2D natural image super-resolution priors. Operating directly on PBR maps, including albedo, roughness, metallic, and normal maps, PBR-SR enables realistic relighting of mesh with SR textures under various lighting conditions.

## Abstract

We present PBR-SR, a novel method for physically based rendering (PBR) texture super resolution (SR). It outputs high-resolution, high-quality PBR textures from low-resolution (LR) PBR input in a zero-shot manner. PBR-SR leverages an off-the-shelf super-resolution model trained on natural images, and iteratively minimizes the deviations between super-resolution priors and differentiable renderings. These enhancements are then back-projected into the PBR map space in a differentiable manner to produce refined, high-resolution textures. To mitigate the effects of view inconsistency and lighting sensitivity inherent to view-based super-resolution, our approach incorporates 2D prior constraints across multi-view renderings, enabling iterative refinement of shared upscaled textures. In parallel, we incorporate identity constraints directly in the PBR texture domain to ensure the upscaled textures remain faithful to the LR input. PBR-SR operates without any additional training

or data requirements, relying entirely on pretrained image priors. We demonstrate that our approach produces high-fidelity PBR textures for both artist-designed and AI-generated LR PBR inputs, outperforming both direct SR models application and prior texture optimization methods. Our results show high-quality outputs in both PBR and rendering evaluations, supporting advanced applications such as relighting.

# 1 Introduction

High-resolution (HR) textures are essential for achieving realistic visuals in physically based rendering (PBR), particularly in applications where close-up details matter, such as in high-fidelity gaming, cinematic effects, and VR experiences. Unfortunately, many existing assets, such as those in older games or legacy 3D models, contain low-resolution textures that result in lower quality render results. Super resolution for PBR textures can significantly enhance the quality of these assets, improving visual clarity and allowing for dynamic relighting and material-aware effects that bring outdated or low-resolution models up to current standards. By directly enhancing low-resolution PBR textures, SR techniques enable better visual fidelity without the need to recreate or replace assets, preserving both artistic intent and compatibility with modern rendering workflows.

Super resolution for PBR textures is particularly difficult due to low- and high-resolution (LR-HR) datasets availability specifically for PBR materials. Unlike natural images, where large-scale, paired datasets exist for supervised SR tasks [5, 6, 23], PBR textures lack such resources, hindering the ability to train models that can accurately upscale these specialized textures. PBR textures require precise alignment across channels (e.g., albedo, roughness, normal) to preserve detail and realism under dynamic lighting and close-up views. However, limited high-quality data hinders the development of effective SR models for such textures.

Existing image SR models can enhance PBR textures by processing every three channels separately. However, current state-of-the-art image SR models are primarily designed for natural images and fail to account for the specific properties of PBR textures [10, 15, 22, 24, 33, 34, 35, 39]. Applying these models directly to PBR textures often yields suboptimal results, as they overlook the spatial coherence and distinct material properties crucial in 3D rendering. An alternative approach is to apply SR models directly to rendered images. However, this introduces view- and lighting-dependent inconsistencies, since these models do not disentangle material properties from appearance. To tackle these challenges, we directly optimize PBR textures with differentiable rendering to a higher resolution. Once refined, these textures can be seamlessly integrated across different environments, maintaining consistency and computational efficiency similar to their low-resolution counterparts.

We propose a zero-shot PBR texture SR method that relies solely on the input LR PBR maps and a pre-trained image SR model. First, we generate an initial SR texture map by combining interpolation-based upsampling with the pre-trained SR model, establishing a strong foundation for further refinement. Second, we apply differentiable PBR rendering on the 3D mesh, synthesizing multi-view renderings from strategically placed camera viewpoints. To enhance high-frequency details, we render images from the same viewpoints and process them with the pre-trained SR model, treating the outputs as pseudo-ground truth (GT) images. We optimize the SR PBR textures by minimizing the discrepancy between differentiable renderings and pseudo-GTs. Leveraging the view-independence of pseudo-GTs, we adopt a robust optimization strategy that jointly updates the target PBR maps and a weighting map for each pseudo-GT, allowing the model to adaptively downweight unreliable supervision. Additionally, to maintain consistency between the SR and LR PBR textures, we introduce PBR consistency constraints that guide view-aware optimization. Extensive experiments validate that our method achieves state-of-the-art performance in both texture fidelity and rendering quality, facilitating applications such as relighting, which are beyond the capabilities of traditional image SR methods.

In summary, our contributions are as follows:

- We present the first zero-shot PBR texture super-resolution framework, effectively leveraging pretrained image SR priors both in UV texture space (for initialization) and in the 2D rendering space (for iterative refinement).

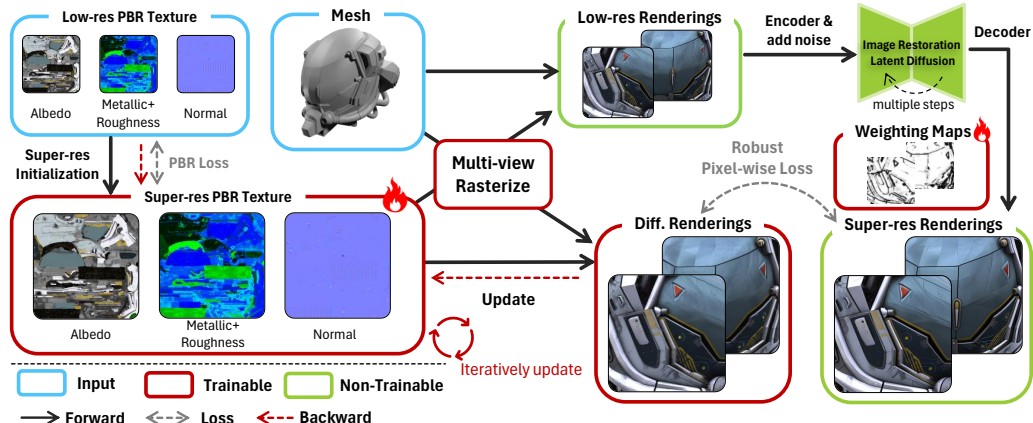

Figure 2: **Pipeline Overview**. PBR-SR begins with a mesh and its LR PBR texture, which are used to initialize the target SR texture (Section 3.2). Renderings are then generated from properly set cameras and passed through an image restoration latent diffusion model to produce SR renderings as pseudo-GT images (Section 3.4). A differentiable mesh rasterizer generates corresponding renderings at the same resolution as SR pseudo-GT images. A robust pixel-wise loss is applied between these renderings and the pseudo-GTs, while a per-view weighting map is jointly optimized to adaptively balance supervision. Additionally, PBR consistency constraints are enforced on the SR textures using the input LR PBR cues. This process iteratively optimizes and refines the SR textures for high-quality results (Section 3.5).

- We introduce an iterative optimization algorithm that refines high-resolution textures by distilling high-frequency details from image priors, while preserving fidelity to the low-resolution input through tailored PBR regularizations.

## 2   Related Work

Realistic materials play a crucial role in generating detailed and realistic images. To this end, many analytical models have been proposed describing how surfaces reflect light. Early works like [7, 29] use simple reflectance models that lack physical principles like energy conservation, which changed with the advent of physically based rendering (PBR) [13, 28]. While PBR has found wide adoption in artist workflows and rendering engines [8, 12, 18], the creation or remastering of spatially varying PBR materials with textures remains a difficult and time-consuming task in which SR methods tailored to materials can help.

**Image Restoration and Super Resolution.**   Recent advances in image restoration and super-resolution (SR) have significantly enhanced image quality across various applications [20, 22, 40, 42, 26]. Notable transformer-based methods, including CAMixerSR [34], HiT-SR [39], and HAT [10], improve reconstruction through hierarchical feature fusion and adaptive feature mixing. MambaIR [15] employs state-space models to address complex spatial relationships effectively. Diffusion-based approaches such as DiffBIR [24] and StableSR [33] excel in generating rich, high-frequency details but typically lack multi-view consistency. Recent works explore multi-view image SR with NeRF-based representations[17, 32, 43] or 3D Gaussian representations[14, 38], enabling view-consistent super-resolution. In this work, we focus on using natural image priors for PBR mesh texture super resolution.

**Mesh Texture Generation.** Recent advancements in mesh texture generation have explored GANs and diffusion models to achieve realistic results [9, 11, 16, 25, 30, 36, 37, 41]. Texurify [31] uses a GAN-based method to generate textures directly on the mesh surface, ensuring consistency across views, while Convolutional Generation of Textured 3D Meshes [27] learns to generate both meshes and textures using 2D supervision. Diffusion models have also proven effective, as seen in Text2Tex [9], which synthesizes textures from text descriptions, and Paint-it [36], which combines deep convolutional optimization with physically based rendering for realistic textures. Decorate3D [16] further enhances text-driven texture generation for real-world applications. These works push the boundaries of quality and realism in texture synthesis for 3D models. Different

from these works that generate texture constrained by category or text prompts, we target using low-resolution texture as important clues and focus on the task of texture map SR.

**Mesh Texture Super Resolution.** Our review of the literature on mesh texture SR yielded only [21], where the texture map contains baked material and lighting. This method finetunes an SR model on a small paired dataset of LR and HR texture maps and applies it directly for texture upscaling. Unlike this approach, we focus on SR for PBR texture maps, enabling broader applications like relighting and material-aware editing. Similarly, Decorate3D [16] uses a standard SR model to upscale generated $512 \times 512$ UV textures to $2048 \times 2048$ but still includes baked material and lighting. In contrast, our method directly optimizes LR PBR texture maps while considering mesh geometry, producing high-quality SR PBR textures that are compatible with artist-created meshes and integrate efficiently with existing PBR frameworks to improve texture fidelity.

## 3 Method

### 3.1 Overview

PBR-SR aims to recover HR PBR texture maps given the LR PBR texture maps with the corresponding mesh, so that the output SR PBR texture maps can produce high-quality renderings. We denote the input LR and the output SR PBR texture maps by albedo $\{\mathbf{T}_{lr}^d, \mathbf{T}_{sr}^d\}$, roughness $\{\mathbf{T}_{lr}^r, \mathbf{T}_{sr}^r\}$, metallic $\{\mathbf{T}_{lr}^m, \mathbf{T}_{sr}^m\}$ and surface normal $\{\mathbf{T}_{lr}^n, \mathbf{T}_{sr}^n\}$, respectively. Once we obtain the SR PBR texture maps, we can produce high-quality renderings under arbitrary lighting and viewpoints.

The overview of PBR-SR is illustrated in Fig. 2. We propose a zero-shot method for PBR texture super-resolution by leveraging pretrained image SR models and differentiable rendering. First, we initialize HR texture maps from the given LR inputs. Next, we render multi-view images and upscale them using a pretrained SR model to generate pseudo-GTs. A differentiable renderer synthesizes images from the same viewpoints using the current HR textures. By iteratively optimizing the textures to minimize the designed loss functions, our method effectively enhances PBR texture details.

### 3.2 PBR Texture Initialization

Given the input of LR PBR texture maps, we use them as the basis to initialize SR PBR maps. Since the albedo map shares the most visual similarity with natural images, the LR albedo map is directly fed to the SR model to produce the SR albedo map $\mathbf{T}_{sr}^d$. For the ARM maps (including ambient occlusion (AO), roughness $\mathbf{T}^r$, and metallic $\mathbf{T}^m$) as well as the normal map $\mathbf{T}^n$, we apply bicubic interpolation to upsample them to the target resolution. If AO is unavailable, an empty map is allocated in the red channel as a placeholder. The initial texture maps will be optimized iteratively through the following modules.

### 3.3 Differentiable Rasterization for PBR

Given the PBR texture maps from Sec. 3.2, we texture the 3D mesh and perform differentiable rasterization to obtain renderings. To render a point $\mathbf{p}$ on the mesh surface, the albedo $\mathbf{a}_\theta \in \mathbb{R}^3$, roughness $\alpha_\theta \in \mathbb{R}$, metallic $m_\theta \in \mathbb{R}$, and normal direction $\mathbf{n}_\theta \in \mathbb{R}^3$ can be queried from the texture maps using UV coordinates. These UV coordinates are either predefined for the corresponding mesh or computed through UV unwrapping. The specular reflectance $F_{0,\theta} \in \mathbb{R}^3$ is calculated as:

$$F_{0,\theta} = 0.04 \cdot (1 - m_\theta) + m_\theta \cdot \mathbf{a}_\theta.$$

The rendered color $L_\theta(\mathbf{p}, \boldsymbol{\omega})$ at surface point $\mathbf{p}$, viewed from direction $\boldsymbol{\omega}$, is computed using the rendering equation:

$$L_\theta(\mathbf{p}, \boldsymbol{\omega}) = \int_{\boldsymbol{\Omega}} L_i(\mathbf{p}, \boldsymbol{\omega}_i) f_\theta(\mathbf{p}, \boldsymbol{\omega}_i, \boldsymbol{\omega}) (\boldsymbol{\omega}_i \cdot \mathbf{n}_\theta) \, d\boldsymbol{\omega}_i, \tag{1}$$

where $\boldsymbol{\omega}_i$ is the incident light direction, $\boldsymbol{\Omega}$ is the hemisphere around the surface normal $\mathbf{n}_\theta$, and $L_i$ is the incident light from the environment map. The BRDF $f_\theta(\mathbf{p}, \boldsymbol{\omega}_i, \boldsymbol{\omega})$ models the material properties, including albedo $\mathbf{a}_\theta$, specular $F_{0,\theta}$, and normal $\mathbf{n}_\theta$.

This rendering equation can be decomposed into a diffuse term $D_\theta(\mathbf{p})$ and a specular term $S_\theta(\mathbf{p}, \boldsymbol{\omega})$ using the Cook-Torrance microfacet model [13]:

$$L_\theta(\mathbf{p}, \boldsymbol{\omega}) = D_\theta(\mathbf{p}) + S_\theta(\mathbf{p}, \boldsymbol{\omega}),$$

$$D_\theta(\mathbf{p}) = \mathbf{a}_\theta(1 - m_\theta)\int_\Omega L_i(\mathbf{p}, \boldsymbol{\omega}_i)(\boldsymbol{\omega}_i \cdot \mathbf{n}_\theta)d\boldsymbol{\omega}_i,$$

$$S_\theta(\mathbf{p}, \boldsymbol{\omega}) = \int_\Omega \frac{D_\theta F_\theta G_\theta}{4(\boldsymbol{\omega} \cdot \mathbf{n}_\theta)(\boldsymbol{\omega}_i \cdot \mathbf{n}_\theta)}L_i(\mathbf{p}, \boldsymbol{\omega}_i)(\boldsymbol{\omega}_i \cdot \mathbf{n}_\theta)d\boldsymbol{\omega}_i, \tag{2}$$

where $D_\theta$, $F_\theta$, and $G_\theta$ represent the microfacet distribution, Fresnel term, and geometric attenuation, respectively. $D_\theta$ and $G_\theta$ depend on roughness $\alpha_\theta$, and $F_\theta$ is based on specularity $F_{0,\theta}$.

Once all surface points are processed, the rendered image $\mathbf{I}_\theta$ is generated from a specific camera. For brevity, we denote the rendering process as $\mathbf{I}_\theta = \mathcal{R}^\mathbf{M}(\mathbf{T}_{sr}^d, \mathbf{T}_{sr}^r, \mathbf{T}_{sr}^m, \mathbf{T}_\theta^n)$, where $\mathcal{R}^\mathbf{M}(\cdot)$ is the differentiable mesh PBR rendering function.

### 3.4 Super Resolution on PBR Rendering

We also use the PBR texture maps from Sec. 3.2 and perform the traditional undifferentiable mesh rendering to render an image from each specified camera viewpoint. Subsequently, we upscale each rendered image $\mathbf{I}_\theta$ with a $4\times$ super-resolution scaling using DiffBIR [24], a unified blind image restoration framework based on diffusion models. DiffBIR consists of two cascaded stages: first, it removes image degradations to yield intermediate high-fidelity restorations; second, it leverages a specially designed IRControlNet module built on latent diffusion models to regenerate realistic high-frequency details. Specifically, we adapt DiffBIR by reducing the denoising steps to five for computational efficiency and modify the text prompts to emphasize PBR rendering characteristics (details provided in Supplementary Materials). The resulting high-resolution image $\mathbf{I}_\theta^{SR}$ serve as pseudo-GT target of each viewpoint for optimizing high-resolution PBR textures.

### 3.5 Iterative Optimization

We use the super-resolution images from Sec. 3.4 to supervise our differentiable renderings from Sec. 3.3 and iteratively optimize the PBR texture maps $\{\mathbf{T}_{sr}^d, \mathbf{T}_{sr}^r, \mathbf{T}_{sr}^m, \mathbf{T}_{sr}^n\}$ initialized in Sec. 3.2. Optimizing PBR texture maps requires rendering the mesh from diverse viewpoints to capture as much surface detail as possible. To achieve this, we normalize the mesh and position cameras on a surrounding sphere to generate a well-distributed set of viewpoints. The camera poses and intrinsics are set to ensure that each rendering captures meaningful surface content across the mesh.

For the initial PBR texture map, we randomly select a batch of $b$ viewpoints in the first iteration. For each view, we render an image using PBR rendering in Sec. 3.4. These rendered images are passed through a pre-trained SR model, which produces the pseudo-GT $\mathbf{I}_\theta^{SR}$.

Simultaneously, at the same view, we render an image $\mathbf{I}^{HR}$ by differentiable rendering in Sec. 3.3 with the same resolution of the pseudo-GT $\mathbf{I}_\theta^{SR}$ using the optimizable PBR texture map $\{\mathbf{T}_{sr}^d, \mathbf{T}_{sr}^r, \mathbf{T}_{sr}^m, \mathbf{T}_{sr}^n\}$. The loss function to update our PBR texture maps consists of two parts: robust pixel-wise loss and PBR constraints.

#### 3.5.1 Robust Pixel-wise Loss

To handle inconsistencies in the pseudo-GT SR images, we propose a robust pixel-wise optimization scheme that downweights unreliable pixels via a learnable per-image pixel weighting map $\mathbf{W}_i(u, v) \in [0, 1]$, where $(u, v)$ denotes a pixel location and $i$ indexes the image. We denote the predicted PBR rendering as $\mathbf{I}_i^{SR} \in \mathbb{R}^{C \times H \times W}$ and the target as $\mathbf{I}_i^{HR}$. The robust pixel-wise loss is defined as:

$$\mathcal{L}_{\text{pix}} = \frac{1}{b}\sum_{i=1}^b \frac{\sum_{u,v} \mathbf{W}_i^2(u, v) \cdot \left\|\mathbf{I}_i^{SR}(u, v) - \mathbf{I}_i^{HR}(u, v)\right\|_2^2}{\sum_{u,v} \mathbf{W}_i^2(u, v)}. \tag{3}$$

This formulation computes a normalized weighted MSE per image and then averages across the batch. Importantly, squaring $\mathbf{W}_i(u, v)^2$ allows sharper modulation of unreliable regions. To regularize the

learned weight maps, we penalize deviation from 1 using a per-pixel mean squared penalty:

$$\mathbf{W} = \sum_{i=1}^{b} \frac{1}{HW} \sum_{u,v} \left(1 - \mathbf{W}_i^2(u,v)\right)^2. \tag{4}$$

The final robust rendering loss becomes:

$$\mathcal{L}_{\text{robust}} = \lambda_{\text{pix}} \cdot \mathcal{L}_{\text{pix}} + \lambda_{\text{reg}} \cdot \mathbf{W}, \tag{5}$$

where $\lambda_{\text{pix}}$ is a global scaling factor and $\lambda_{\text{reg}}$ is the regularization weight. This design ensures that gradient flow is preserved across all pixels while adaptively reducing the influence of uncertain regions (e.g., view inconsistency and shadows), and aims to stabilize PBR learning under noisy pseudo-ground-truth supervision.

### 3.5.2   PBR Constraints

**PBR Consistency Loss:** This term ensures consistency between the optimized HR PBR texture maps and the LR texture input across all material properties. It encourages the refined HR textures to preserve the structure and material integrity of the original inputs while allowing for higher-resolution details:

$$\mathcal{L}_{\text{pbr}} = \sum_{\mathbf{T} \in \mathbf{T}^d, \mathbf{T}^r, \mathbf{T}^m, \mathbf{T}^n} w_{\mathbf{T}} \left(\|\text{Pool}(\mathbf{T}_\theta) - \mathbf{T}_{\text{lr}}\|_1 + \lambda_{\text{ssim}} \cdot \mathcal{L}_{\text{SSIM}}(\text{Pool}(\mathbf{T}_\theta), \mathbf{T}_{\text{lr}})\right), \tag{6}$$

where $\mathbf{T}_\theta$ denotes the current HR PBR texture map being optimized; $\mathbf{T}_{\text{lr}}$ indicates the corresponding LR textures, and $w_{\mathbf{T}}$ denotes the weight of the L1 loss applied between textures. $\text{Pool}(\cdot)$ is the average pooling operator with a kernel size equal to the PBR upscaling factor, which downsamples the HR texture to match the LR input. $\mathcal{L}_{\text{ssim}}$ is a SSIM loss and $\lambda_{\text{ssim}}$ is a weighting factor.

**PBR Total Variation (TV) Regularization:** To encourage spatial smoothness and suppress undesirable artifacts in the optimized PBR textures, we introduce a total variation loss:

$$\mathcal{L}_{\text{tv}} = \sum_{\mathbf{T} \in \{\mathbf{T}_{sr}^d, \mathbf{T}_{sr}^r, \mathbf{T}_{sr}^m, \mathbf{T}_{sr}^n\}} \left(\|\nabla_x \mathbf{T}\|_1 + \|\nabla_y \mathbf{T}\|_1\right), \tag{7}$$

where $\nabla_x$ and $\nabla_y$ represent the horizontal and vertical gradients of the texture maps, respectively. Minimizing the TV loss promotes smooth, artifact-less textures while preserving important structural details.

The total loss for the optimization is then formulated as:

$$\mathcal{L}_{\text{total}} = \mathcal{L}_{\text{robust}} + \lambda_{pbr} \mathcal{L}_{\text{pbr}} + \lambda_{tv} \mathcal{L}_{\text{tv}},$$

where $\lambda_{pbr}$ and $\lambda_{tv}$ are weighting factors to balance these three loss terms.

## 4   Results

In this section, we present the experimental setup, evaluation metrics, and results obtained from our proposed method for PBR texture super resolution.

### 4.1   Experimental Setup

**Datasets.** Since there is no established benchmark on mesh PBR texture super resolution, we collect a set of PBR meshes with rich texture information [1, 2, 4] (***Artist-designed Dataset***). This collection contains 16 different high-quality meshes with high-resolution PBR maps in $4096 \times 4096$ or $8192 \times 8192$ resolutions, and their low-resolution counterparts with a $4\times$ smaller resolution. In addition, we evaluate on 32 AI-generated PBR-textured meshes sourced from the Hyper3D commercial platform [3] (***AI-generated Dataset***). These meshes contain generated PBR textures with greater variety and realism, and each PBR channel is provided at $512 \times 512$ resolution. We apply $4\times$ super-resolution to generate $2048 \times 2048$ outputs. Each model contains a set of texture maps that include albedo, metallic, roughness, and normal maps. We use low-resolution meshes as the input to the SR model and the high-resolution counterparts as ground truth for performance evaluation.

Table 1: Quantitative comparison of PBR texture maps and renderings PSNR from $\times 4$ PBR SR results on the Artist-designed Dataset. $\dagger$ indicates a supervised method finetuned on PBR data. $*$ refers an optimization-based method. The red and orange respectively denote the best and the second-best results.

| Method | Albedo | Roughness | Metallic | Normal | Renderings |
|---|---|---|---|---|---|
| OSEDiff [35] | 23.495 | 24.095 | 26.922 | 23.957 | 23.804 |
| DiffBIR [24] | 24.842 | 27.563 | 27.493 | 24.743 | 25.495 |
| SwinIR [22] | 26.990 | 26.241 | 28.564 | 23.410 | 24.952 |
| HAT [10] | 25.260 | 30.559 | 30.536 | 23.561 | 26.205 |
| StableSR [33] | 25.398 | 27.648 | 28.953 | 24.191 | 25.489 |
| CAMixerSR [34] | 26.297 | 28.273 | 30.473 | 24.056 | 26.791 |
| CAMixerSR-FT $\dagger$ | 27.800 | 30.642 | 28.961 | 25.655 | 27.928 |
| Paint-it SR [36] $*$ | 25.682 | 29.729 | 28.955 | 28.237 | 23.959 |
| PBR-SR (**Ours**) | 29.731 | 31.602 | 31.889 | 29.088 | 28.001 |

**Pretrained SR Models.** For albedo initialization and generating pseudo-GT (Sec . 3.4, Sec . 3.5), we utilize DiffBIR[24], a two-stage diffusion-based model that efficiently restores details through degradation removal and detail regeneration. All pre-trained models remain fixed throughout the optimization process, making our framework universally applicable to future models.

**Implementation Details.** Our implementation leverages the differentiable renderer from [19] to produce rendered images from selected viewpoints. These images are then super-resolved using the same image SR model to create pseudo-GTs for optimization. Unless explicitly specified, the same environment lighting setup is used for both optimization and evaluation. The optimization uses the Adam optimizer with a constant learning rate of $1 \times 10^{-4}$. In each iteration, we use a batch size of 4, which corresponds to 4 viewpoints. We stop the iterative optimization after 2000 iterations.

## 4.2 Comparison with Baselines

We compare our method with several alternative techniques for PBR texture SR. We evaluate the performance against:

- **Image SR Methods**: Including SwinIR [22], HAT [10], CAMixerSR [34], OSEDiff [35], StableSR [33] and DiffBIR [24], which are applied directly on the PBR texture maps.

- **CAMixerSR-FT**: Since CAMixerSR achieves overall balanced performance on all PBR channels, we use its architecture and pretrained weights from natural images, and then fine-tune it on a collection of 24,000 LR-HR PBR map pairs ($120 \times 120$ to $480 \times 480$) comprising diffuse, ARM, and normal maps extracted from the Poly Haven website [4] (no overlap with the evaluation data). Unlike the other baselines and our PBR-SR, CAMixerSR-FT is a supervised baseline.

- **Paint-it SR**: Paint-it [36] is a model designed for text-driven mesh PBR texture generation. We adopt a variant of Paint-it as an optimization-based baseline for the PBR texture SR task by replacing its text-to-image diffusion model and SDS loss with an image super-resolution model and a render loss function.

**Evaluation Metrics.** We evaluate the performance of our PBR texture SR method in both PBR maps and renderings. We assess the quality of the PBR texture maps on albedo, roughness, metallic, and normal maps individually by comparing the PSNR (Peak Signal-to-Noise Ratio) of SR results with ground truth high-resolution PBR maps. A higher PNSR value is better. In addition, we compare the rendering quality from novel views (views not used during the optimization process), utilizing PSNR to quantify the difference between renderings from SR PBR results and the ground truth PBR maps.

**Quantitative Evaluation.** Table 1 presents a quantitative comparison of various super-resolution methods on PBR maps and final renderings, measured by PSNR. The evaluated maps include albedo, roughness, metallic, and normal, alongside the resulting rendered images. The proposed method, PBR-SR, is compared against both supervised baseline (e.g., CAMixerSR-FT), optimization-based approaches (Paint-it SR), and state-of-the-art transformer and diffusion-based methods. The proposed PBR-SR consistently outperforms all baselines, achieving the highest scores across all five metrics. While CAMixerSR-FT, a supervised method fine-tuned on PBR data, performs well on Roughness and rendering metrics, it still lags behind PBR-SR. Paint-it SR shows strong results on the normal map

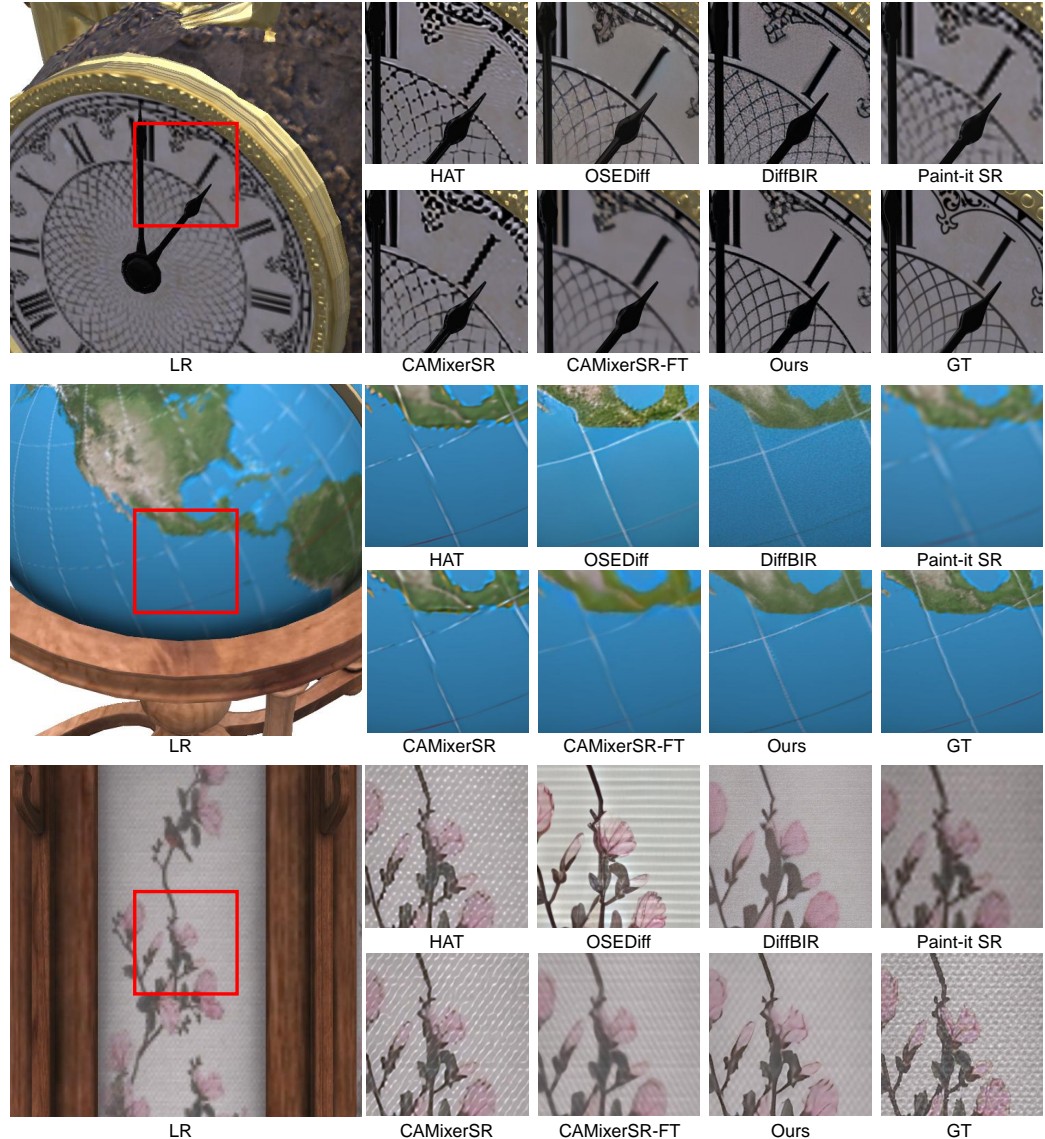

Figure 3: Comparison of renderings from PBR texture SR results. Our method produces consistently higher-quality renderings with improved detail.

but falls short on others, especially in rendering quality. Transformer-based models like SwinIR and HAT perform competitively in specific channels but struggle with consistent performance across all maps. Overall, PBR-SR delivers the most balanced and superior results, highlighting its effectiveness in high-quality material and renderings. In Table 2, we provide quantitative results comparing our method against the two strongest baselines. The trends observed on the AI-generated meshes are consistent with those on the original 16 high-quality test cases. Across both evaluation sets, our method achieves robust improvements in all PBR texture channels and final rendered appearance, further confirming its generalization ability across diverse mesh sources and content types.

**Qualitative Comparison on Renderings.** In addition to quantitative metrics, we present qualitative comparisons of rendered images using our optimized PBR textures versus those produced by baseline methods. Fig. 3 shows renderings under identical lighting conditions, featuring the *Table Clock*, the *Cradle Globe*, and the *Chinese Chandelier* at $4\times$ SR. The LR and GT display the renderings from the LR PBR and GT PBR textures. HAT and CAMixerSR cannot reveal accuracy in all PBR channels, leading to distortion or reflection artifacts. OSEDiff, which employs a diffusion model, produces high-frequency details, though it often diverges from the LR cues, erasing or altering LR

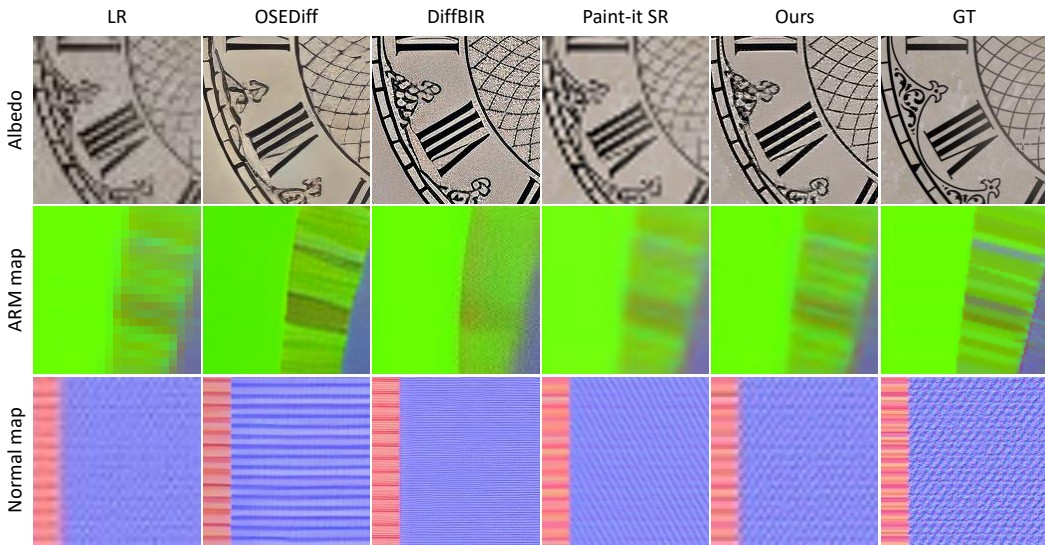

Figure 4: Comparison of PBR tiles from $4\times$ PBR SR results of different meshes from our test set. Our method significantly improves the LR PBR on all channels and outperforms the inference-based and optimization-based baselines.

Table 2: Quantitative comparison of PBR texture maps and renderings PSNR from $\times 4$ PBR SR results on the AI-generated Dataset. [†] indicates a supervised method finetuned on PBR data. The red and orange respectively denote the best and the second-best results.

| Method | Albedo | Roughness | Metallic | Normal | Renderings |
|---|---|---|---|---|---|
| HAT | 26.368 | 32.750 | 31.794 | 28.235 | 28.312 |
| CAMixerSR-FT [†] | 31.439 | 33.195 | 27.583 | 31.054 | 30.841 |
| PBR-SR (**Ours**) | 33.841 | 35.059 | 35.352 | 34.564 | 31.208 |

Table 3: Ablation study on our PBR mesh collection (Artist-designed Dataset). PSNR of PBR map channels and resulting renderings are reported.

| Setting | Albedo | Roughness | Metallic | Normal | Renderings |
|---|---|---|---|---|---|
| w/o PBR Loss | 26.116 | 27.661 | 29.220 | 27.130 | 26.572 |
| w/o PBR TV Regularization | 27.929 | 30.744 | 31.492 | 28.507 | 27.593 |
| w/o Robut Pixel-wise Loss | 28.084 | 30.817 | 31.734 | 28.639 | 27.666 |
| full | 29.731 | 31.602 | 31.889 | 29.088 | 28.001 |

details, with material property shifts causing incorrect brightness and shading. DiffBIR introduces too much noise and struggles to get accurate, detailed renderings. CAMixerSR-FT learns to produce plausible outputs in the texture space through supervision on training data. However, this does not guarantee high-quality results in the rendering space. In practice, it often leads to distortions or a lack of sharpness in the rendered appearance, as the model is not explicitly optimized for rendering fidelity. Paint-it SR, an optimization-based model, iteratively improves PBR renderings but appears more blurred than ours. Our method achieves results that are closest to the GT renderings.

**Qualitative Comparison on PBR Texture Maps.** Fig. 4 visualizes PBR texture tiles, with three rows from top to bottom showing the albedo map, the ARM map (with channels for ambient occlusion, roughness, and metallic), and the surface normal map. From left to right, the columns correspond to LR PBR texture, PBR texture generated by OSEDiff, DiffBIR, Paint-it SR, and Our method PBR-SR, GT HR PBR texture. As shown in the figure, our method significantly outperforms all inference- and optimization-based baselines. Especially in comparison with the LR PBR texture, our method demonstrates a substantial improvement in texture quality, highlighting the effectiveness of applying image SR priors to the task of PBR texture SR.

Table 4: Ablation on the image SR model used in our PBR-SR pipeline. We list the quantitative comparison of PBR texture maps and renderings PSNR from $\times 4$ PBR SR results.

| Method | Albedo | Roughness | Metallic | Normal | Renderings |
|---|---|---|---|---|---|
| CAMixerSR | 27.793 | 29.927 | 31.784 | 27.294 | 27.466 |
| StableSR | 27.678 | 31.025 | 32.258 | 27.995 | 26.415 |
| DiffBIR | 29.731 | 31.602 | 31.889 | 29.088 | 28.001 |

**Ablation Study.** Table 3 presents an ablation study assessing the contribution of key components in our method. Removing the PBR loss leads to the most significant drop in PSNR across all metrics, particularly in albedo and renderings, underscoring its importance. Excluding TV regularization or the robust pixel-wise loss also results in consistent performance drops, especially in roughness and normal maps. The full model achieves the highest PSNR in all categories, confirming that each component contributes to the overall quality, with the most notable gains in PBR quality and rendering fidelity.

**Ablation on Image SR Models in PBR-SR.** In Table 4, we compare three models used in our pipeline for both initialization and rendering super resolution, and report the final optimized SR results of PBR maps and corresponding renderings on the Artist-designed Dataset. DiffBIR achieves the best performance across most PBR channels, including Albedo, Roughness, and Normal, as well as in the final rendering PSNR. Specifically, it outperforms CAMixerSR and StableSR by a notable margin in both texture fidelity and rendered appearance, indicating its superior ability to preserve structural details and material realism in the super-resolved outputs. While StableSR performs well on the Metallic channel, it underperforms on others and results in lower rendering quality overall. Moreover, DiffBIR is computationally more efficient than StableSR, offering faster inference while maintaining high-quality outputs. This makes it a favorable choice for integration into our iterative optimization pipeline, where both accuracy and runtime efficiency are critical.

**Limitations.** While PBR-SR performs well on PBR texture SR tasks, it struggles with severely degraded LR textures due to the limitations of natural image priors. Multimodal cues like text or sketches could help in such cases. The current zero-shot optimization removes the need for training data but is relatively slow, limiting real-time applicability. Future work will explore faster optimization and stronger PBR-specific priors with multimodal integration.

For additional details and results, please refer to the Supplementary Materials.

## 5   Conclusion

We have introduced PBR-SR, the first zero-shot super-resolution method specifically designed for enhancing PBR textures on 3D meshes. By integrating pre-trained image SR models with differentiable mesh rendering, our approach effectively restores detailed textures from low-quality inputs while ensuring accurate preservation of material consistency. PBR-SR significantly improves texture fidelity and rendering quality, outperforming existing state-of-the-art image SR models and optimization-based baselines, without requiring explicit training data. By effectively preserving and enhancing material properties, our method supports downstream applications such as realistic relighting, facilitating more detailed and visually compelling 3D scenes. We believe this work provides valuable insights into leveraging natural image priors for PBR texture restoration, paving the way toward specialized PBR priors in future 3D content generation.

**Acknowledgements.** This work is funded by the ERC Consolidator Grant Gen3D (101171131), the German Research Foundation (DFG) Research Unit "Learning and Simulation in Visual Computing". Additionally, we would like to thank Angela Dai for the video voice-over.

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

In this supplementary material, we provide additional details and results that are not included in the main paper due to space constraints. The attached video offers a brief overview of our method, along with qualitative results demonstrating the performance of PBR-SR.

# A  Implementation Details

## A.1  Datasets

**Artist-designed Dataset.** In the main paper, we evaluate quantitative results on a collection of PBR meshes sourced from [1, 2, 4]. To better evaluate PBR texture super-resolution performance, we select meshes that contain rich and diverse textures in different PBR channels. Table 5 lists the meshes and their corresponding website links. For evaluation, we treat the downloaded high-resolution textures as ground truth. If paired low-resolution textures are provided, we use them directly as input to our method. Otherwise, we generate low-resolution inputs by applying a Gaussian blur followed by bicubic downsampling on the high-resolution textures. Our usage fully complies with the Royalty Free License terms, as the models are only used internally for testing and are not redistributed in any digital or physical form. The dataset was obtained under its original license and is not redistributed here. We used the dataset solely for method evaluation. No training, fine-tuning, or derivative model development was performed using this data.

Table 5: List of 16 Artist-designed meshes used in our evaluation. Each URL links to the corresponding mesh page.

| No. | Model Name | URL |
|---|---|---|
| 1 | Chinese Chandelier | https://polyhaven.com/a/chinese_chandelier |
| 2 | Corset | https://github.com/KhronosGroup/glTF-Sample-Assets/tree/main/Models/Corset |
| 3 | Damaged Helmet | https://github.com/KhronosGroup/glTF-Sample-Models/tree/main/2.0/DamagedHelmet |
| 4 | Shoulder Strap | https://www.cgtrader.com/free-3d-models/sports/equipment/shoulder-strap |
| 5 | Globe | https://www.cgtrader.com/free-3d-models/interior/interior-office/globe-59b6052d-2afb-4ee5-81c5-02d3d50223cc |
| 6 | Under Armour Volleyball Shoe | https://www.cgtrader.com/free-3d-models/character/clothing/under-armour-womens-hovr-highlight-ace-volleyball-shoe |
| 7 | Table Clock | https://www.cgtrader.com/free-3d-models/furniture/tableware/table-clock-eb2f344f-abb4-4439-803d-d080edb7742f |
| 8 | Medieval Chest | https://www.cgtrader.com/free-3d-models/furniture/other/medieval-chest-with-ornaments |
| 9 | Cradle Globe | https://www.cgtrader.com/free-3d-models/interior/interior-office/cradle-globe |
| 10 | Armored Man | https://www.cgtrader.com/3d-models/character/sci-fi-character/armored-man-low-poly-game-ready-model |
| 11 | Lounge Chair 1 | https://www.cgtrader.com/free-3d-models/interior/living-room/lounge-chair-3d-model-low-poly-pbr-textures |
| 12 | Lounge Chair 2 | https://www.cgtrader.com/free-3d-models/interior/living-room/lounge-chair-3d-model-low-poly-pbr-textures |
| 13 | Old Table | https://www.cgtrader.com/free-3d-models/household/other/old-chair-old-stool-and-old-table |
| 14 | Old Stool | https://www.cgtrader.com/free-3d-models/household/other/old-chair-old-stool-and-old-table |
| 15 | Old Chair | https://www.cgtrader.com/free-3d-models/household/other/old-chair-old-stool-and-old-table |
| 16 | Vintage Cozy Rocking Chair | https://www.cgtrader.com/free-3d-models/furniture/chair/vintage-cozy-rocking-chair |

**AI-generated Dataset.** To strengthen the evaluation, we add 32 additional meshes from the commercial Hyper3D website [3], which features AI-generated PBR textures with diverse content. Table 6 lists the meshes and their corresponding website links. Each sample includes PBR maps at $512 \times 512$ resolution as input and $2048 \times 2048$ resolution as ground truth (GT). We perform $4\times$ super-resolution as described in the main paper, keeping all experimental settings consistent with those used for the Artist-Designed meshes. The dataset was obtained under its original license and is not redistributed here.

## A.2  Optimization

The mesh is normalized to a unit size during the optimization process. The camera is positioned at a distance of 3.25 units with a field of view (FoV) of 10 degrees. We use 750 views in total, sampled from 15 different elevations, with 50 views evenly distributed at each elevation. For rendering evaluation, we use 240 views from 6 elevations, with 40 views sampled from each elevation, ensuring an even distribution across the scene. During optimization, the $\lambda_{\text{pix}}$ is set to 100 and $\lambda_{\text{reg}}$ is 0.5 in the robust pixel-wise loss term. The weighting map is optimized in resolution of $64 \times 64$ and is interpolated to $\mathbf{W}_i$ with the same resolution as the rendering's image when calculating $\mathcal{L}_{\text{pix}}$. We assign different weights to the channels of the PBR texture maps in the PBR consistency loss function: we set the weight to 1.0 for the diffuse $w_{\mathbf{K}^d}$, roughness $w_{\mathbf{K}^r}$, and normal map $w_{\mathbf{K}^n}$, and 0.1 for the metallic map $w_{\mathbf{K}^m}$. The SSIM part is weighted by $\lambda_{\text{ssim}} = 10$ for all. The overall PBR consistency loss is given a weight $\lambda_{pbr} = 10$ and the PBR TV loss is weighted by $\lambda_{tv} = 0.5$. The optimization process runs for 2000 iterations. On a single NVIDIA A6000 RTX GPU, optimizing a mesh with PBR textures takes around 30 minutes with 2K-to-8K resolution and less than 8 minutes with 1K-to-2K

Table 6: List of 32 AI-generated meshes from the Hyper3D website used in our evaluation. Each URL links to the corresponding mesh page.

| No. | Model Name | URL |
| --- | --- | --- |
| 1 | 51a273a2-e71a-4050-b6eb-87b7dce6907b | https://hyper3d.ai/rodin/51a273a2-e71a-4050-b6eb-87b7dce6907b |
| 2 | 8bdfe26e-dcac-4bb5-85e1-76a3120248f8 | https://hyper3d.ai/rodin/8bdfe26e-dcac-4bb5-85e1-76a3120248f8 |
| 3 | 7ba15124-8c21-4421-a43c-8098a62e4b6a | https://hyper3d.ai/rodin/7ba15124-8c21-4421-a43c-8098a62e4b6a |
| 4 | 3c9e3ff4-7793-4ec6-825e-8f3a5793eb3d | https://hyper3d.ai/rodin/3c9e3ff4-7793-4ec6-825e-8f3a5793eb3d |
| 5 | 2a6d3eeb-309f-4a6d-89e4-4dcaeb0ac781 | https://hyper3d.ai/rodin/2a6d3eeb-309f-4a6d-89e4-4dcaeb0ac781 |
| 6 | 62bbcaba-d5bc-4a29-9191-a500c74306f2 | https://hyper3d.ai/rodin/62bbcaba-d5bc-4a29-9191-a500c74306f2 |
| 7 | 453fe7bc-a49c-42d7-aab0-d0a4360ed8df | https://hyper3d.ai/rodin/453fe7bc-a49c-42d7-aab0-d0a4360ed8df |
| 8 | 00792bae-57fe-49c4-88ea-8e885038ea62 | https://hyper3d.ai/rodin/00792bae-57fe-49c4-88ea-8e885038ea62 |
| 9 | 282954ca-f348-4ba9-b5ad-a1d142b230f3 | https://hyper3d.ai/rodin/282954ca-f348-4ba9-b5ad-a1d142b230f3 |
| 10 | b70a199e-a309-4197-971f-d093262bce7d | https://hyper3d.ai/rodin/b70a199e-a309-4197-971f-d093262bce7d |
| 11 | c8c1776c-4f12-4e9b-aa2c-9a72b7dcb699 | https://hyper3d.ai/rodin/c8c1776c-4f12-4e9b-aa2c-9a72b7dcb699 |
| 12 | c26cd539-7d46-4b6f-ad55-bf0f28ffb33a | https://hyper3d.ai/rodin/c26cd539-7d46-4b6f-ad55-bf0f28ffb33a |
| 13 | d78eb744-7b74-47b6-8121-8c3b75829ab3 | https://hyper3d.ai/rodin/d78eb744-7b74-47b6-8121-8c3b75829ab3 |
| 14 | fc2a26ae-a469-47a1-a3f9-f80c4693ecff | https://hyper3d.ai/rodin/fc2a26ae-a469-47a1-a3f9-f80c4693ecff |
| 15 | fe2318d4-96b7-4886-8838-5106f1391d0f | https://hyper3d.ai/rodin/fe2318d4-96b7-4886-8838-5106f1391d0f |
| 16 | 5c419b6b-f6b9-4be3-be03-6034bdbdcbe6 | https://hyper3d.ai/rodin/5c419b6b-f6b9-4be3-be03-6034bdbdcbe6 |
| 17 | 17a4fd66-cba7-43af-ae63-1b85bb08db14 | https://hyper3d.ai/rodin/17a4fd66-cba7-43af-ae63-1b85bb08db14 |
| 18 | 70892277-56ea-4b93-b62c-cfa7c8ebb1ae | https://hyper3d.ai/rodin/70892277-56ea-4b93-b62c-cfa7c8ebb1ae |
| 19 | b9e2bf1a-d48f-4bd1-aac1-2e10db4b6abe | https://hyper3d.ai/rodin/b9e2bf1a-d48f-4bd1-aac1-2e10db4b6abe |
| 20 | b4254276-6acd-405f-b68c-98262d394564 | https://hyper3d.ai/rodin/b4254276-6acd-405f-b68c-98262d394564 |
| 21 | f440f7f4-ef0d-4ad1-ad6c-aaeba5e13088 | https://hyper3d.ai/rodin/f440f7f4-ef0d-4ad1-ad6c-aaeba5e13088 |
| 22 | fc19a47e-15f8-4faf-86ec-953ddae28ebc | https://hyper3d.ai/rodin/fc19a47e-15f8-4faf-86ec-953ddae28ebc |
| 23 | 1e997330-88d4-4c28-81d8-f58b22c0c137 | https://hyper3d.ai/rodin/1e997330-88d4-4c28-81d8-f58b22c0c137 |
| 24 | 4ee35b93-3134-492d-944b-892608bcb55f | https://hyper3d.ai/rodin/4ee35b93-3134-492d-944b-892608bcb55f |
| 25 | 19419b03-c0d6-4297-b783-bad4df29ce77 | https://hyper3d.ai/rodin/19419b03-c0d6-4297-b783-bad4df29ce77 |
| 26 | 56395d6a-88e9-40d4-a72d-d31dbee8377d | https://hyper3d.ai/rodin/56395d6a-88e9-40d4-a72d-d31dbee8377d |
| 27 | 65617646-4fe3-457e-948b-6066ad04765d | https://hyper3d.ai/rodin/65617646-4fe3-457e-948b-6066ad04765d |
| 28 | aef2393f-1889-4404-a563-6890cca9743c | https://hyper3d.ai/rodin/aef2393f-1889-4404-a563-6890cca9743c |
| 29 | b613c088-addb-493f-ad42-9383fa148081 | https://hyper3d.ai/rodin/b613c088-addb-493f-ad42-9383fa148081 |
| 30 | ceae5972-860d-41f6-92c5-27ff35d1a2d9 | https://hyper3d.ai/rodin/ceae5972-860d-41f6-92c5-27ff35d1a2d9 |
| 31 | e49e49f5-c3f0-4a85-9f1a-1d264be6edf4 | https://hyper3d.ai/rodin/e49e49f5-c3f0-4a85-9f1a-1d264be6edf4 |
| 32 | f61cfdb3-d6a4-480a-b504-3aa81074a187 | https://hyper3d.ai/rodin/f61cfdb3-d6a4-480a-b504-3aa81074a187 |

resolution offline on average. The optimizable parameters are limited to the high-resolution PBR textures, as we directly update them using computed gradients.

## A.3 Baselines

We leverage the official implementations of SwinIR [22], HAT [10], CAMixerSR [34], StableSR [33], OSEDiff [35] and DiffBIR [24] as baselines. For $N$ times PBR texture super resolution, we initialize SR PBR texture maps using the corresponding models trained for specific $N$ times super resolution, if the model is available with their pretrained weights.

For Paint-it SR, we adopt the core deep convolutional architecture and optimization framework of Paint-it, but substantially modify it to suit the super-resolution (SR) task. Specifically, we replace the original text-to-image diffusion model and Score Distillation Sampling (SDS) loss with an image super-resolution model and a pixel-wise render loss. The deep convolutional block is re-initialized to produce zero output initially, enabling it to learn only residual components over the interpolated low-resolution (LR) PBR maps. This residual learning setup allows the model to refine high-frequency details while preserving the low-frequency information already captured in the LR inputs. Although Paint-it SR shares a similar optimization framework with the original method, it is explicitly re-purposed for SR rather than text-to-texture synthesis. These modifications make it a strong baseline, yet our proposed approach surpasses it through a more effective integration of SR priors and view-consistent optimization.

For CAMixerSR-FT, we make it a supervised baseline using PBR textures to fine-tune the off-the-shelf CAMixerSR weights pre-trained on natural images. We use 19 high-quality PBR meshes from the Poly Haven website [4] (no overlap with the evaluation data), each containing diffuse, ARM (ambient occlusion, roughness, metallic), and normal maps at a resolution of $4096 \times 4096$. For training data generation, all texture maps are downsampled to $1024 \times 1024$ using bicubic interpolation. From these, we randomly extract $480 \times 480$ patches as high-resolution targets and their corresponding $120 \times 120$

Table 7: Ablation on robust pixel loss in our PBR-SR pipeline. We list the quantitative comparison of PBR texture maps and renderings PSNR from $\times 4$ PBR SR results.

| Method | Albedo | Roughness | Metallic | Normal | Renderings |
|---|---|---|---|---|---|
| w/o robust | 27.748 | 30.199 | 31.826 | 27.702 | 27.477 |
| Ours | 29.731 | 31.602 | 31.889 | 29.088 | 28.001 |

Table 8: Ablation on the rendering resolution of pseudo-GT used in our PBR-SR pipeline. We list the quantitative comparison of PBR texture maps and renderings PSNR from $\times 4$ PBR SR results. Results of $4\times$ SR on the *Table Clock* mesh are reported.

| Resolution | Albedo | Roughness | Metallic | Normal | Renderings |
|---|---|---|---|---|---|
| 128 | 31.913 | 30.814 | 33.375 | 34.284 | 22.485 |
| 256 | 32.549 | 30.737 | 34.013 | 34.181 | 23.841 |
| 512 | 33.476 | 30.646 | 34.085 | 34.900 | 24.659 |
| 768 | 34.318 | 30.857 | 35.803 | 35.229 | 24.555 |
| 1024 | 36.077 | 30.644 | 36.296 | 35.620 | 24.856 |

downsampled versions as low-resolution inputs. This process yields a total of 24,000 LR-HR texture pairs, which are used to fine-tune the $\times 4$ PBR super-resolution model. We fine-tune the pretrained CAMixerSR [34] model on each set (diffuse, ARM, or normal). The model is initialized from the official checkpoint and trained for 50K iterations using the AdamW optimizer with a learning rate of $1 \times 10^{-4}$, weight decay of $1 \times 10^{-5}$, and $(\beta_1, \beta_2) = (0.9, 0.99)$. A multi-step learning rate scheduler is used with decay milestones at 20K and 40K iterations and a decay factor of 0.5. We adopt an L1 loss in the texture (UV) space with equal weighting and no learning rate warmup. The training is conducted with exponential moving average (EMA) enabled, using a decay rate of 0.999. All experiments are performed with strict weight loading from the pretrained model, and the best checkpoint is selected based on validation performance. The entire training process takes approximately 9.5 hours.

# B  Additional Results

**Impact of Robust Pixel-wise Loss.** In the main paper, we adopt a robust pixel-wise loss to mitigate the impact of artifacts in pseudo-GT renderings, such as view inconsistency, lighting variation, and shadow misalignment. To validate the necessity of this design, we conduct an ablation by replacing our robust loss with a standard pixel-wise rendering loss: $\mathcal{L}_{\text{pix}} = \frac{1}{b} \sum_{i=1}^{b} \left\| \mathbf{I}_\theta^{SR} - \mathbf{I}_\theta^{HR} \right\|_2^2$, where $b$ denotes the number of rendered views and $\theta$ represents the optimizable parameters (i.e., the super-resolved PBR maps). This baseline assumes uniform confidence across all pixels and ignores image-specific supervision noise. As shown in Fig. 5, using this naive loss often leads to overfitting in unreliable regions such as specular highlights or shadows, resulting in artifacts and inconsistent textures. In contrast, our robust formulation introduces a per-image pixel-wise weight map $\mathbf{W}_i(u, v) \in [0, 1]$ that adaptively downweights uncertain pixels. We regularize these weights toward one to ensure stable optimization: $\mathcal{L}_{\text{robust}} = \lambda_{\text{pix}} \cdot \mathcal{L}_{\text{pix}} + \lambda_{\text{reg}} \cdot \mathcal{R}(\mathbf{W})$. This design improves both the stability and quality of optimization, particularly in regions prone to multi-view inconsistencies. As shown in Table 7, our full model with robust pixel-wise loss achieves consistently higher PSNR across PBR channels and final renderings compared to the baseline using the standard pixel-wise rendering loss on the Artist-designed Dataset. These gains highlight the benefit of adaptively downweighting unreliable pixels during optimization. Fig. 5 presents a comparison of renderings under different lighting conditions from our PBR-SR method and its variant without the robust pixel-wise loss. Without the robust loss, optimization relies on a uniform pixel-wise average over multi-view supervision, which often introduces blocky artifacts. This issue is exacerbated by the PBR consistency loss, which operates in texture space using average pooling and implicitly assumes equal reliability across all pixels. In contrast, our robust loss adaptively downweights unreliable regions (e.g., shadows or view-specific lighting inconsistencies), resulting in cleaner and more physically plausible texture reconstructions.

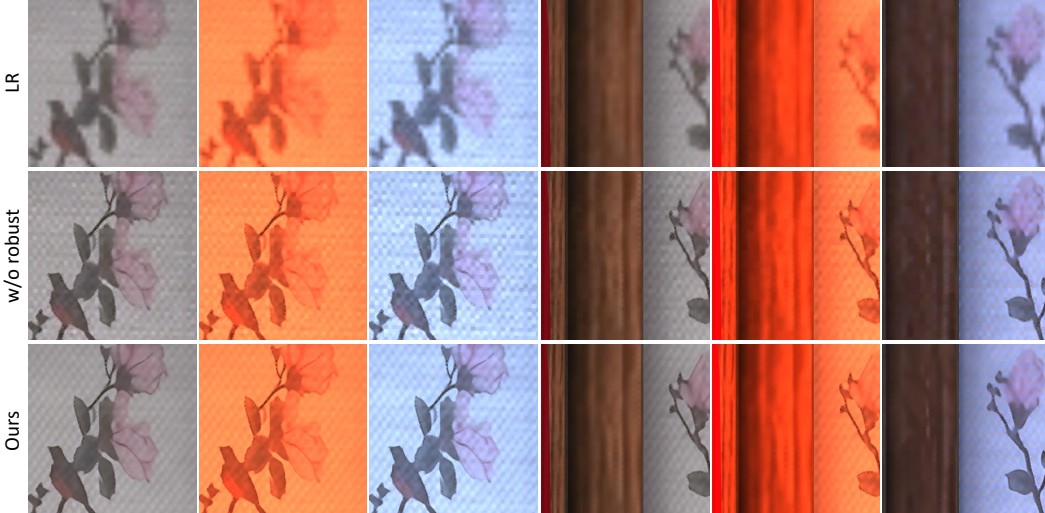

Figure 5: Comparison of renderings under different lighting from our PBR-SR and its variant without using robust pixel-wise loss. Both methods significantly improve the rendering quality from the LR PBR texture, while ours achieves high-fidelity by getting sharper and more natural details.

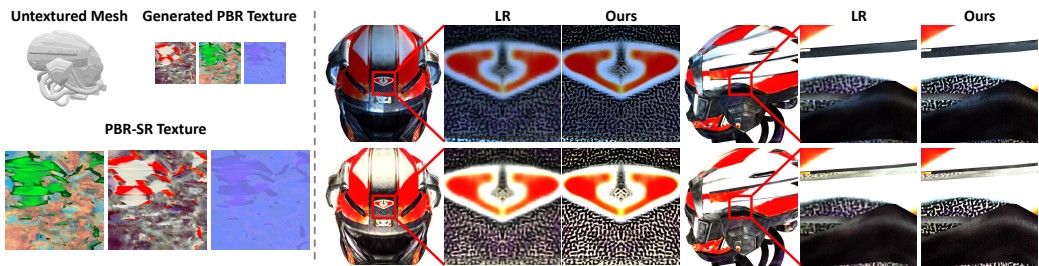

Figure 6: Texture SR for generated PBR texture. Left: we adopt Paint-it [36] to generate LR texture and use PBR-SR for ×4 SR. Right: we compare renderings from LR and Ours from two viewpoints. The two rows are under different lighting conditions.

**Ablation on Rendering Resolution in Optimization.** We assess the impact of rendering resolution on optimization performance using the *Table Clock* model at $128 \times 128$, $256 \times 256$, $512 \times 512$, $768 \times 768$, and $1024 \times 1024$. As shown in Table 8, increasing the resolution of the pseudo-ground truth renderings consistently improves performance across most channels. Notably, the Albedo and Metallic maps benefit the most as the resolution increases. Rendering quality also improves steadily, with PSNR rising from 22.49 to 24.86, indicating that higher-resolution renderings provide more accurate supervision signals during optimization. The Normal map shows consistent gains, reflecting enhanced geometric fidelity, while the Roughness channel exhibits only minor variation, suggesting lower sensitivity to resolution changes. Given the diminishing returns beyond $1024 \times 1024$ and the significantly increased computational and memory costs at higher resolutions, we adopt $1024 \times 1024$ as the default resolution for DiffBIR outputs in our main experiments.

**Clarification on Robust Pixel-wise Loss (Weighting Map Resolution)..** As shown in Table 3 of the main manuscript, we have verified the overall effectiveness of our robust pixel-wise loss. To further analyze the impact of weighting map resolution, we conducted experiments on the *ShoulderStrap* mesh using different weighting map sizes ranging from 8 to 256. The results are summarized in Table 9. We observe that performance improves steadily from 8 to 64, with diminishing returns beyond 128. This indicates that moderate-resolution maps (e.g., 64 or 128) are sufficient to capture spatial inconsistencies introduced by the SR model, while higher resolutions offer negligible gains but incur significant memory overhead, as a separate weighting map must be optimized for each rendering view. Therefore, we adopt a resolution of 64 as a practical trade-off between reconstruction quality and computational efficiency.

Table 9: Impact of weighting map resolution on reconstruction performance (PSNR) for the *Shoulder-Strap* mesh. Moderate resolutions (64–128) achieve a good balance between quality and efficiency.

| Weight Map Res. | Albedo | Roughness | Metallic | Normal | Renderings |
|---|---|---|---|---|---|
| 8 | 34.543 | 32.501 | 36.272 | 26.455 | 27.838 |
| 16 | 34.641 | 32.551 | 35.477 | 26.685 | 27.923 |
| 32 | 34.743 | 32.588 | 35.198 | 26.575 | 28.063 |
| 64 | 34.758 | 32.615 | 35.264 | 25.983 | 28.163 |
| 128 | 34.770 | 32.649 | 35.193 | 25.766 | 28.448 |
| 256 | 34.765 | 32.663 | 35.214 | 26.594 | 28.338 |

Table 10: Comparison of interpolation methods for initializing PBR texture maps on the *Globe* mesh.

| Metallic/Roughness Init | Normal Init | Albedo | Roughness | Metallic | Normal | Renderings |
|---|---|---|---|---|---|---|
| bicubic | bicubic | 35.638 | 32.002 | 35.343 | 34.299 | 36.240 |
| bicubic | bicubic + normalize | 35.638 | 32.033 | 35.341 | 32.855 | 36.243 |
| nearest | bicubic | 35.647 | 31.269 | 32.756 | 34.301 | 35.883 |

**Interpolation Method for PBR Maps.** To evaluate the impact of different interpolation strategies for initializing metallic, roughness, and normal maps, we conducted an ablation study on the *Globe* mesh. The results are summarized in Table 10. Bicubic interpolation for metallic and roughness slightly outperforms nearest-neighbor interpolation, providing smoother transitions that facilitate optimization—even for maps with binary-valued regions. For normal maps, bicubic interpolation without normalization yields better performance (PSNR 34.30 vs. 32.86). We attribute this to compression and quantization artifacts commonly present in publicly available normal maps, which result in vector norms slightly below one; post-interpolation normalization amplifies these artifacts and degrades quality. Overall, bicubic interpolation serves as a robust and effective initialization strategy, while any minor inconsistencies are further corrected through our optimization process.

**Robustness Analysis on Degraded LR Textures.** To evaluate the robustness of our optimization-based method under degraded inputs, we conducted additional experiments on the *TableClock* mesh using two common types of degradation applied to the LR albedo map: (i) Gaussian noise with standard deviations $\sigma = \{5, 10, 15, 20\}$ (in the 0–255 range), and (ii) JPEG compression with quality factors $\{80, 60, 40, 20, 10\}$, where lower values indicate stronger compression. For each setting, we report both the final PSNR after optimization and the PSNR improvement ($\Delta$) relative to initialization, for the albedo map and the rendered appearance (averaged over views). Results are summarized in Table 11. Our method consistently improves both albedo and rendering quality, even under moderate or severe degradation. Performance gradually decreases as degradation increases, which is expected; however, the optimization process continues to yield meaningful improvements over the initialization. Notably, our approach remains relatively robust to JPEG compression, maintaining stable performance even at high compression levels. These findings confirm that while input quality influences the final outcome, our method can tolerate a reasonable range of degradation and still benefit from the optimization process.

**Texture SR for Generated PBR Texture.** We adopt Paint-it [36] to generate $1024 \times 1024$ PBR textures from an untextured mesh (*Damaged Helmet*), and apply PBR-SR to perform $4\times$ super-resolution on the generated PBR maps. As shown in Fig. 6 (right), our method successfully introduces high-frequency details that are absent in the low-resolution renderings. However, the performance of PBR-SR on generated textures is inherently limited by the quality of the input. Since the generated textures often lack fine-grained structural cues, it becomes challenging for the SR model to hallucinate or infer additional detail. Moreover, current PBR texture generation methods struggle to produce material properties that are physically consistent and visually comparable to those crafted by professional artists. We believe that future advances in generative PBR modeling could complement PBR-SR effectively. A more realistic and physically plausible generation of initial textures would allow PBR-SR to further enhance detail quality, potentially approaching the fidelity of artist-created PBR assets.

Table 11: Robustness evaluation on degraded LR albedo inputs for the *TableClock* mesh. The results report PSNR after optimization and the corresponding improvement ($\Delta$) from initialization.

| Degradation | Albedo PSNR | Renderings PSNR | Albedo $\Delta$PSNR | Renderings $\Delta$PSNR |
|---|---|---|---|---|
| None | 33.904 | 24.429 | +8.041 | +1.031 |
| Gaussian 5 | 32.608 | 24.223 | +7.080 | +0.989 |
| Gaussian 10 | 30.199 | 24.085 | +4.332 | +0.748 |
| Gaussian 15 | 27.913 | 23.826 | +2.062 | +0.517 |
| Gaussian 20 | 26.174 | 23.560 | +0.325 | +0.322 |
| JPEG 80 | 33.554 | 24.282 | +6.873 | +0.719 |
| JPEG 60 | 33.292 | 24.352 | +6.325 | +0.677 |
| JPEG 40 | 32.979 | 24.323 | +6.092 | +0.648 |
| JPEG 20 | 32.138 | 23.853 | +6.278 | +0.828 |
| JPEG 10 | 30.974 | 22.978 | +6.006 | +0.887 |

**Qualitative Results on Composed Scenes.** To evaluate PBR-SR on scenes with multiple objects, we composed eight meshes from the CGTrader dataset. Figure 7 compares scene renderings using the input low-resolution (LR) textures and our PBR-SR textures. Our method significantly enhances the details of each object in the scene. Additionally, we compare scene renderings under different environment lighting conditions in Figure 8. We tested three environment maps from Poly Haven [4]: *Billiard Hall*, *De Balie*, and *Beach Parking*. The results demonstrate that PBR-SR consistently improves rendering quality across all lighting scenarios.

**Discussion on Learning-based and Optimization-based Methods for PBR Texture SR.** While the learning-based baseline (CAMixerSR finetuned on our PBR dataset) achieves moderate improvements, its performance is limited by the scarcity of large-scale, diverse PBR texture–mesh datasets and by image-space loss functions that fail to capture physically based effects such as relighting consistency and material fidelity. In contrast, our optimization-based approach is data-efficient and directly minimizes rendering-space error through differentiable rendering, leading to superior visual realism and robustness under novel lighting, albeit with slower optimization. We view these two paradigms as complementary: learning-based models provide efficient initialization and inference, whereas optimization-based refinement ensures physically grounded fidelity. Future research could explore hybrid frameworks that integrate differentiable rendering losses and PBR-specific regularization into the training of feed-forward networks, combining the strengths of both approaches for high-quality, physically consistent PBR texture super resolution.

**Failure Cases.** Our model can generally generate HR PBR textures based on the provided LR PBR inputs. The main failure cases we observed occur when the input LR PBR maps are of extremely poor quality, lacking sufficient structural or textural clues. In such scenarios, the model struggles to infer plausible high-frequency details, which leads to suboptimal outputs.

**Video.** We include a supplementary video [1] showcasing various aspects of our method. The video provides an overview of the approach, a visualization comparing the LR PBR texture maps with the PBR-SR outputs, and results under different relighting conditions. Additionally, it features comparisons with baseline methods and renderings of a composed scene, highlighting the differences between LR textures and PBR-SR textures.

---

[1] https://youtu.be/eaM5S3Mt1RM

Rendering with LR PBR Texture          Rendering with Our texture

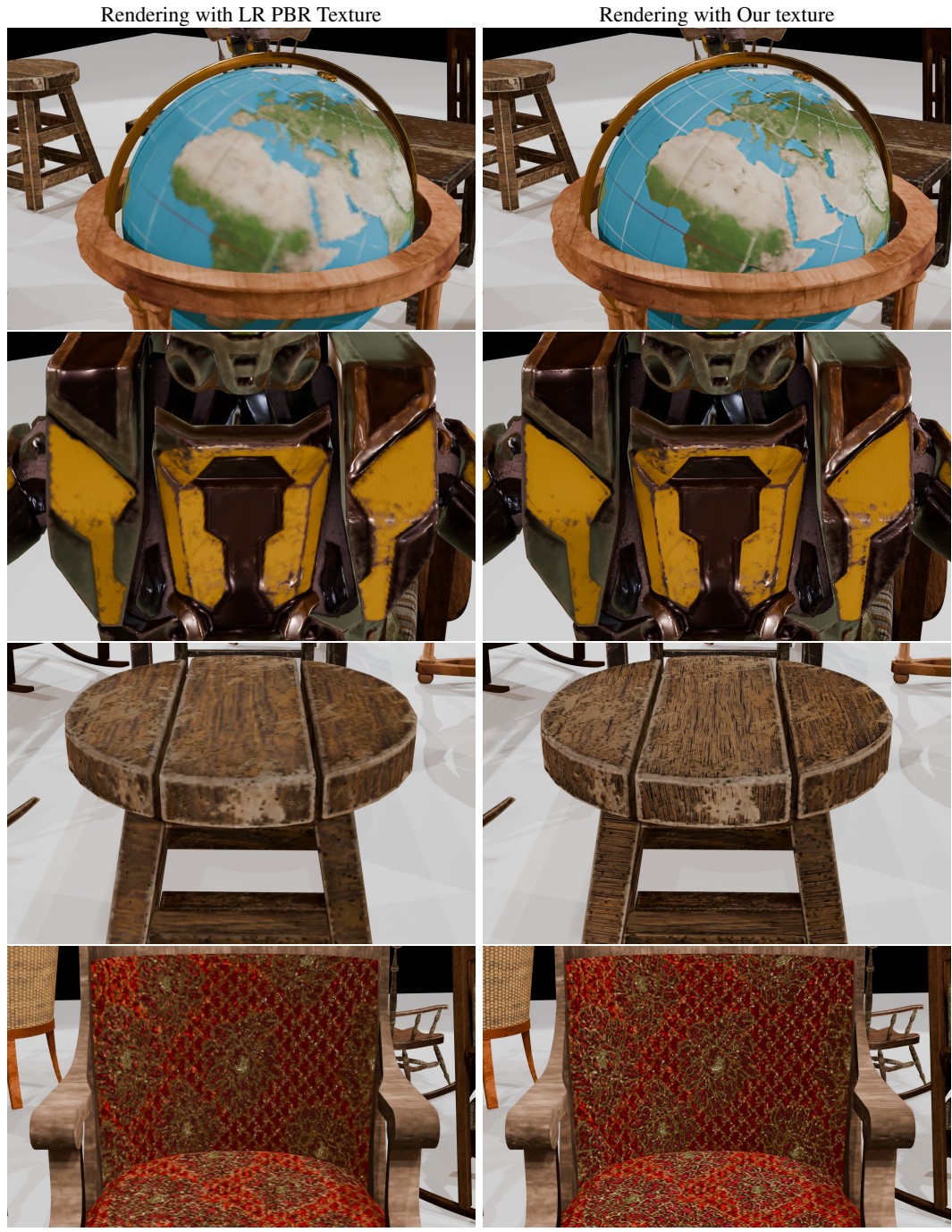

Figure 7: Comparison of scene renderings from LR textures and PBR-SR textures.

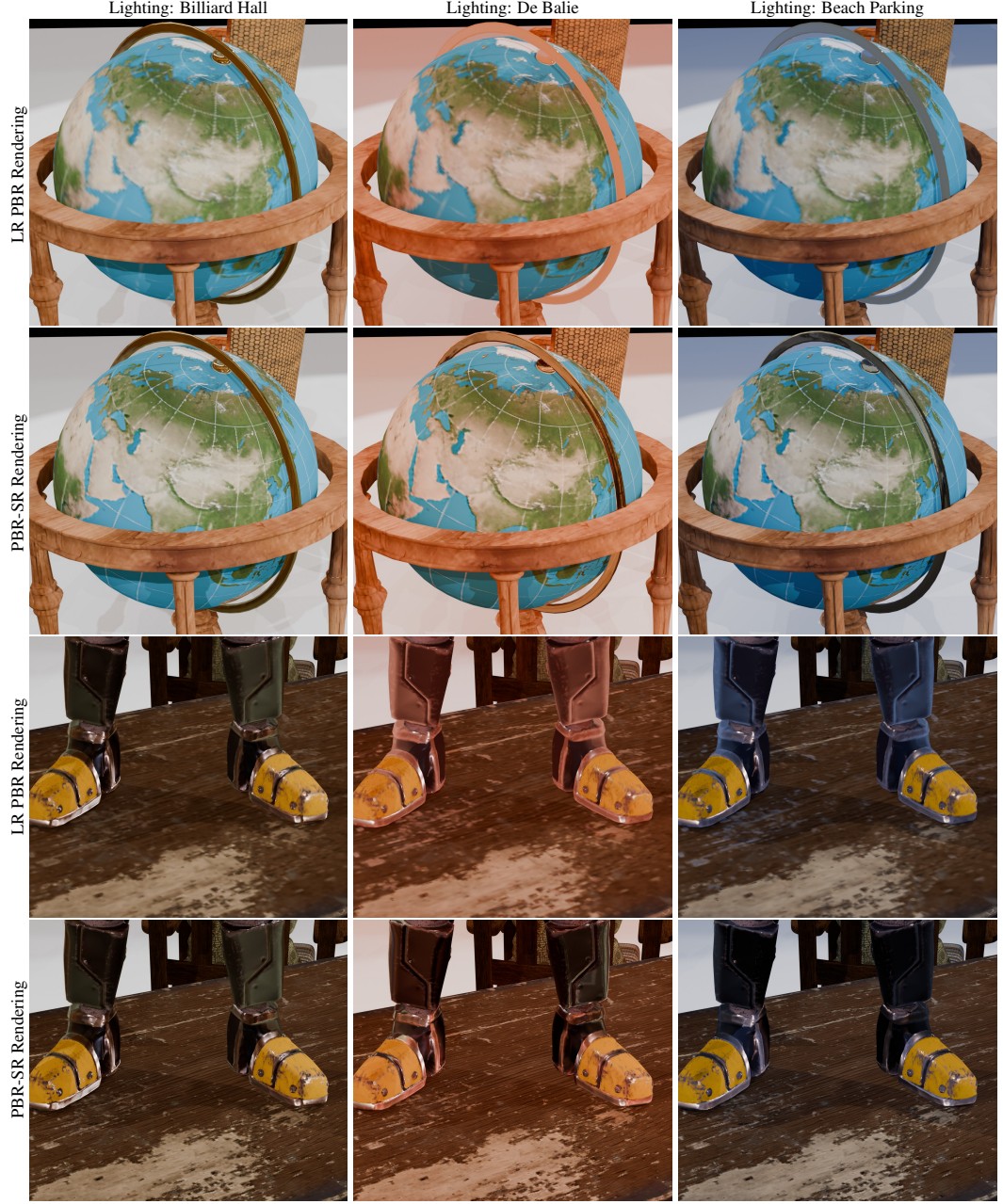

Figure 8: Comparison of scene renderings from LR textures and PBR-SR textures under novel environment lighting.

