# OpenReview forum: "PBR-SR: Mesh PBR Texture Super Resolution from 2D Image Priors"
_NeurIPS.cc/2025/Conference — NeurIPS 2025 poster_

### Official Review · Reviewer_h7Gq · 2025-06-23

**Clarity:** 4
**Significance:** 4
**Originality:** 4
**Rating:** 5
**Confidence:** 3

**Summary:**

This paper presents PBR-SR, a super-resolution framework for PBR textures. The core idea is to reconstruct high-resolution PBR texture maps using a pre-trained diffusion prior combined with differentiable rendering-based supervision. PBR-SR leverages an off-the-shelf super-resolution model trained on natural images, and iteratively minimizes the deviations between the diffusion prior outputs and differentiable renderings. Experimental results demonstrate strong performance, although some issues deserve further discussion.

**Questions:**

Please refer to the Weakness section. I do not consider an additional experiment strictly necessary, but I believe a thoughtful discussion of the trade-offs between learning- and optimization-based inference is important. If the authors provide additional experimental evidence, I would consider increasing my score.

**Ethical Concerns:**

["NO or VERY MINOR ethics concerns only"]

**Final Justification:**

I have read the rebuttal and appreciate the authors’ clarifications. They have addressed my main concerns, and I am increasing my score from 4 to 5.

**Limitations:**

yes

**Quality:**

3

**Strengths And Weaknesses:**

Strengths:
1. The paper addresses a practical and less explored problem: super-resolution of PBR texture maps on 3D meshes, while maintaining fidelity under relighting. The motivation is clear and relevant to realistic asset generation workflows.
2. The proposed framework combines a pre-trained diffusion prior with differentiable rendering-based iterative optimization. Along with the use of PBR loss, TV regularization, and pixel-wise supervision, the method significantly improves performance on PBR super-resolution, as shown in comprehensive experiments.

Weaknesses:
1. The proposed method relies on an optimization-based inference pipeline, which requires per-input optimization and can be relatively slow at test time, limiting its practicality in real-world deployment.
While CAMixerSR is fine-tuned on PBR textures as a learning-based baseline and does achieve improved results, the performance gain is relatively modest compared to the proposed optimization-based approach. However, the paper does not analyze whether this reflects a fundamental limitation of learning-based methods in this setting, or whether it is due to other factors such as loss function design.
A more in-depth discussion (or experiments, if possible) would help clarify the trade-offs between learning- and optimization-based approaches in terms of performance and inference efficiency.

---

> ### Author Rebuttal · Authors · 2025-07-31
>
> We sincerely thank the reviewer for the constructive feedback. We are glad that our work was recognized as addressing a 'practical and less explored problem' with 'clear motivation' and that our combination of diffusion priors, differentiable rendering, and pixel-wise supervision was found to 'significantly improve performance'. Below, we provide detailed responses to the concerns raised.
>
> **1. Optimization-based inference pipeline is relatively slow**
>
> The optimization-based inference pipeline in PBR-SR can be slower compared to fully feed-forward models. This design choice is motivated by our aim to achieve the highest possible fidelity under novel lighting conditions and get rid of extra PBR data reliance, which current learning-based models struggle to generalize to due to domain shift and limited training data. While this makes PBR-SR better suited for offline content creation (e.g., game asset preparation or digital twin generation), real-time deployment could benefit from more efficient alternatives.
>
> **2. Limitation of learning-based methods, or if the design can be improved**
>
> While our current learning-based baseline (CAMixerSR fine-tuned on our PBR dataset) achieves only modest improvements, we do not attribute this to an inherent limitation of learning-based methods. Rather, we believe that two key factors limit their current performance:
> - Data limitation: The lack of large-scale, diverse PBR texture datasets paired with 3D mesh geometry under varying lighting conditions makes it difficult for learning-based methods to generalize effectively. Our current fine-tuning is performed on relatively limited PBR data, which likely restricts performance.
> - Loss design: Existing loss functions in image space (e.g., pixel-wise or perceptual losses) do not adequately capture physically-based rendering characteristics, such as relighting consistency or material-specific shading effects, especially under novel views or illumination.
>
> Looking ahead, we believe our optimization-based framework provides valuable insights for improving learning-based methods. Specifically, it opens up the possibility of incorporating physically motivated constraints, such as differentiable rendering loss and PBR-specific regularization, into the training pipeline.
> For example, when only PBR textures are available, 2D supervision can be used as a proxy. When PBR texture-mesh pairs are available, our differentiable rendering loss can be integrated into the training loop to guide learning with richer, geometry-aware supervision.
> We see this as a promising future direction to combine the efficiency of feed-forward inference with the fidelity of physically-grounded supervision.
>
> **3. Trade-offs between learning- and optimization-based approaches**
>
> Learning-based methods (e.g., CAMixerSR-FT) offer fast inference but rely on large-scale paired data and are typically supervised in texture space. As discussed in Lines 255–268, this can lead to distortions or a lack of sharpness in rendered appearance under novel lighting, since rendering fidelity is not explicitly optimized.
>
> In contrast, our optimization-based approach is data-efficient and directly minimizes rendering-space error via differentiable rendering, leading to better relighting consistency, though at the cost of slower optimization.
>
> As shown in Table 1, our method outperforms CAMixerSR-FT across all PBR channels on tested artist-created meshes:
>
> - Albedo: 27.800 → 29.731
> - Roughness: 30.642 → 31.602
> - Metallic: 28.961 → 31.889
> - Normal: 25.655 → 29.088
>
> These results confirm the advantage of rendering-aware supervision. Figure 3 further shows that while CAMixerSR-FT produces plausible textures, it often fails to preserve material fidelity under relighting, unlike our method.
>
> Similarly, we additionally tested on 32 PBR meshes that are AI-generated, and our method still outperforms CAMixerSR-FT across all PBR channels:
> - Albedo: 31.439 → 33.821
> - Roughness: 33.195 → 35.059
> - Metallic: 27.583 → 35.352
> - Normal: 31.054 → 34.564
>
>
> We emphasize that our work fills a current gap in this space by proposing an optimization framework that leverages 2D priors while ensuring high rendering fidelity, even without additional PBR training data. At the same time, we believe future work could explore hybrid strategies, such as:
> - Using learning-based models (e.g., CAMixerSR) for efficient initialization;
> - Incorporating rendering-aware loss functions into the training of feed-forward networks;
> - Jointly training on both texture-only and mesh-texture paired data, using our differentiable supervision as a guiding signal.
>
> We have expanded our discussion accordingly in the revised manuscript.

---

> > ### Comment · Reviewer_h7Gq · 2025-08-01
> >
> > Thank you for the response. I have read the rebuttal and the other reviews. As the authors have addressed my main concerns, I am inclined to revise my score from 4 to 5 based on the current state of the submission.

---

> > > ### Author Response · Authors · 2025-08-05
> > >
> > > Thank you very much for your response and helpful feedback. Please don’t hesitate to let us know if you have any further questions or if we can clarify anything else.

---

### Official Review · Reviewer_MDkT · 2025-06-25

**Clarity:** 3
**Significance:** 3
**Originality:** 3
**Rating:** 4
**Confidence:** 4

**Summary:**

This paper presents PBR-SR, an optimization-based method for mesh PBR texture super resolution.
It is the first paper to consider the task of PBR texture super resolution, which is important in game and multi-media industry.
It leverages the 2D priors from image domain super resolution model and distill the knowledge into UV domain SR task.
The main contribution of the paper is a zero-shot texture super resolution framework, and an optimization method with proposed losses.
Good quality SR results are shown in the experiment part.

**Questions:**

1. More details should be discussed about the mechanism of the weight map.
2. More cases should be shown.
3. The implementation of the baseline method Paint-it SR should be clarified.

**Ethical Concerns:**

["NO or VERY MINOR ethics concerns only"]

**Final Justification:**

The response has addressed my concerns.

**Limitations:**

Yes

**Quality:**

2

**Strengths And Weaknesses:**

Strength

- The paper identifies the critical industrial demand for PBR texture super-resolution and, for the first time, proposes a solution to address this challenge.
- A novel distillation approach is proposed to transfer knowledge from the natural image domain to the UV domain, enabling zero-shot capability in a training-free manner. This effectively solves the problem of the lack of LR-HR PBR texture data pairs.
- The paper is well written, and the structure is well organized and easy to follow.
- The presented method is simple to implement in real-world applications.
- Good super-resolution results are shown, proving its superiority.

Weaknesses

- The mechanism of the weight map (the robust pixel-wise loss) seems unclear to the reviewer. The reviewer understand the additional solution space introduced can reduce the influence of the inconsistency produced by the SR model. However, the 64x64 resolution is not enough for such inconsistency. More detail should be discussed.
- The choice of the light condition during rendering should be clarified since it is important for optimizing ARM and normal maps.
- In the evaluation part, only 16 cases are used for comparison. This number is too small to prove the method's generalizability. The reviewer would like to see more cases. Also, since the authors claim that the method can process AI-generated meshes, it would be better to show more cases from that domain.
- The impact of different natural image super-resolution models on the final results has not been discussed. Does the method rely on DiffBIR or any other SR method that can be plug-and-use (which would be a great advantage of the method)?
- The implementation of the baseline method Paint-it SR is not clearly described. It seems to use a similar optimization framework to the proposed method and only replaces the optimization target with a U-Net, which produces the PBR residue. I understand that in the original Paint-it paper, neural reparameterization is used for frequency scheduling. However, in the SR task, low-frequency information is already known (LR texture), and it seems redundant to add another U-Net residue. The authors should discuss the main difference between it and the proposed method.

---

> ### Author Rebuttal · Authors · 2025-07-31
>
> We appreciate the detailed and valuable feedback from the reviewer, and we are glad that our method was recognized as addressing a 'critical industrial demand' with a 'novel distillation approach'. We also appreciate that our work was found to be 'well written', 'easy to follow', and 'simple to implement', while demonstrating 'good super-resolution results'. We respond to each of the points raised below.
>
> **1. Clarification on robust pixel-wise loss (weighting map resolution)**
>
> As shown in Table 2 of the main manuscript, we verify the overall effectiveness of this loss. To further analyze the impact of resolution, we conducted experiments using different weighting map sizes (8 to 256)  on the *ShoulderStrap* mesh. The results are summarized below:
>
> | Weight Map Res. | Albedo | Roughness | Metallic | Normal | Renderings |
> |------------|------------|---------------|--------------|------------|-------------|
> | 8                   | 34.543     | 32.501        | 36.272       | 26.455     | 27.838                |
> | 16                  | 34.641     | 32.551        | 35.477       | 26.685     | 27.923                |
> | 32                  | 34.743     | 32.588        | 35.198       | 26.575     | 28.063                |
> | 64                  | 34.758     | 32.615        | 35.264       | 25.983     | 28.163                |
> | 128                 | 34.770     | 32.649        | 35.193       | 25.766     | 28.448                |
> | 256                 | 34.765     | 32.663        | 35.214       | 26.594     | 28.338                |
>
> We observe that performance improves steadily from 8 to 64, with diminishing returns beyond 128. This suggests that moderate-resolution maps (e.g., 64 or 128) are sufficient to capture useful spatial inconsistencies introduced by our SR model. Higher resolutions offer marginal gains but incur significantly more memory overhead, as a separate weight map is optimized and stored for each rendering view. Hence, we adopt a resolution of 64 as a practical balance between quality and efficiency. These clarifications have been added to the revised manuscript.
>
> **2. Choice of lighting conditions**
>
> We utilize a single environment map in our experiments, as stated in the manuscript (line 220). Further experiments with varying and multiple environment maps demonstrated that the results were robust under moderate lighting changes, primarily due to constraints imposed by LR PBR maps, rendering the method insensitive to variations in illumination during optimization. This clarification has been added to the manuscript.
>
> **3. Evaluation scale and AI-generated meshes**
>
> To address the reviewer’s concern and further validate generalizability, we have added 32 AI-generated PBR-textured meshes sourced from the Hyper3D commercial platform. These meshes contain procedurally or AI-generated PBR textures with greater variety and realism, and each PBR channel is provided at 512×512 resolution. We apply 4× super-resolution to generate 2048×2048 outputs. The experimental protocol remains consistent with that used in the main 16-mesh evaluation.
> We present preliminary results comparing our method with the strongest baselines. Final results will be included in the revised manuscript, along with comparisons to other baselines and additional quantitative comparisons.
>
> | Method           | Albedo | Roughness | Metallic | Normal | Renderings |
> |------------------|--------|-----------|----------|--------|------------------|
> | HAT              | 26.368 | 32.750    | 31.794   | 28.235 | 28.312         |
> | CAMixerSR-FT     | 31.439 | 33.195    | 27.583   | 31.054 | 30.841           |
> | PBR-SR (ours)    | 33.841 | 35.059    | 35.352   | 34.564 | 31.208           |
>
> These additional evaluations on AI-generated PBR textures reinforce the general applicability and robustness of our method, beyond handcrafted or artist-designed assets.
>
> **4. Impact of different super-resolution models**
>
> We provide the analysis of the impact of different SR models in Appendix Section B *Ablation on Image SR Models in PBR-SR*, along with Table 4, where we compare three different SR models (DiffBIR, CAMixerSR, and StableSR) used for both initialization and rendering-time SR.
>
> Our results show that DiffBIR consistently achieves the best performance across most PBR channels, including Albedo, Roughness, and Normal, as well as in final rendering PSNR. It outperforms CAMixerSR and StableSR by a notable margin in terms of both texture fidelity and perceptual rendering quality. While StableSR performs well on the Metallic channel, it underperforms on others and leads to lower overall rendering quality.
>
> Importantly, our method is not tied to DiffBIR—it supports plug-and-play integration of other SR models. This flexibility is a key strength of our pipeline. However, we adopt DiffBIR as the default due to its strong balance of performance and efficiency: it not only yields high-quality outputs but also offers faster inference than StableSR, which is especially valuable in our iterative optimization process.
> We have clarified this point in the revised manuscript to emphasize the generality of our method with respect to different SR backbones.
>
> **5. Implementation details of  Paint-it SR baseline**
>
> In Section 4.2 and Appendix Section A.3, we retain the core convolutional architecture and optimization framework of Paint-it, but make the following key changes:
> - We replace the original text-to-image diffusion model and Score Distillation Sampling (SDS) loss with an image super-resolution model and a pixel-wise render loss.
> - The deep convolutional block is re-initialized to output zero initially, allowing it to learn only the residual components over interpolated LR PBR maps.
> - This residual learning setup allows the model to refine details while preserving the low-frequency information already present in the LR maps.
> While Paint-it SR shares a similar optimization framework, it is explicitly re-purposed for SR rather than text-to-texture synthesis. These modifications make it a strong baseline, yet our proposed method still outperforms it due to a more effective integration of SR priors and view-consistent optimization.
>
> We have clarified the above points in the revised manuscript.

---

> > ### Comment · Reviewer_MDkT · 2025-08-05
> >
> > Thanks for your response. They largely solve my concerns. I will raise my rating to borderline accept.

---

> > > ### Author Response · Authors · 2025-08-05
> > >
> > > Thank you very much for your response and helpful feedback. Please don’t hesitate to let us know if you have any further questions or if we can clarify anything else.

---

### Official Review · Reviewer_45eH · 2025-07-01

**Clarity:** 2
**Significance:** 2
**Originality:** 2
**Rating:** 4
**Confidence:** 4

**Summary:**

This paper presents PBR-SR, a zero-shot method for generating high-resolution PBR texture maps on 3D meshes. The approach addresses the challenging problem of enhancing low-resolution PBR textures by leveraging pretrained 2D image super-resolution models without requiring additional training data. The method works by initializing high-resolution textures from low-resolution inputs, generating multi-view renderings that are processed through a pretrained SR model to create pseudo GT images, and then iteratively optimizing the PBR textures using differentiable rendering to minimize discrepancies between rendered images and pseudo-GTs.

The paper's main contributions are:
- A robust pixel-wise loss with learnable weighting maps to handle view inconsistencies.
- PBR consistency constraints to preserve material properties.
- A comprehensive evaluation demonstrating good performance over direct application of image SR methods and optimization-based baselines.

**Questions:**

Please refer to weaknesses for more details.

**Ethical Concerns:**

["NO or VERY MINOR ethics concerns only"]

**Final Justification:**

Thank you to the authors for their comprehensive efforts in addressing my concerns during the review process. The additional clarifications and explanations have significantly improved my understanding of the work, particularly regarding the robustness analysis on degraded LR textures and the detailed discussion of failure modes and limitations.

Based on these substantial improvements in technical clarity, I am raising my score for this submission. I strongly recommend that the authors incorporate these detailed explanations into the main paper rather than keeping them solely in the rebuttal responses. Specifically, the robustness analysis on degraded LR textures and the comprehensive failure mode analysis would greatly benefit readers' comprehension of the technical contributions and practical considerations.

**Limitations:**

While the authors do include a limitations paragraph, it is quite brief and doesn't adequately address several important limitations. More discussion should be included:

- More detailed analysis of computational requirements and scalability limitations.
- Discussion on failure cases/modes.
- Discussion on the method's dependence on the pretrained 2D SR model and how the the method performs across different material types.

**Quality:**

2

**Strengths And Weaknesses:**

### Strengths
- The paper is well-written and easy to follow.
- The paper presents a well-motivated method that bridges 2D image priors and 3D PBR texture enhancement. The use of differentiable rendering combined with iterative optimization is reasonable. The pixel-wise loss with learnable weighting maps is a practical solution to handle view inconsistencies. The PBR consistency constraints effectively preserve material properties while allowing detail improvement.
- The method achieves better results comparing against multiple strong baselines. The ablation study clearly demonstrates the contribution of each component.

### Weaknesses
- The evaluation is conducted on only 16 meshes, which represents a small test set for validating method's generalization. While the authors mention the data are collected from multiple sources, the diversity in terms of material types, geometric complexity, and texture details should be discussed. Besides, it is strongly recommended to evaluate the method on more comprehensive benchmarks.
- The method requires iterative optimization with 2000 iterations and multiple view renderings per iteration which the authors acknowledge as "relatively slow." However, the paper lacks detailed computational analysis. Lack of timing comparisons and memory usage with respect to mesh complexity and texture resolution makes it difficult to understand computational costs for practical usage.
- The method adopts "zero-shot" paradigm. Such paradigm is typically sensitive to input quality. However, the paper doesn't provide analysis of how the method performs when LR textures exhibit various forms of degradation. It is strongly recommended to add robustness experiments by adding different types of noise to LR textures and analyzing how they affect model performances.
- The method replies on pretrained SR models, which may limit its effectiveness on PBR textures that differ from natural images. The paper doesn't evaluate how this dependency affects performance across different material types.

---

> ### Author Rebuttal · Authors · 2025-07-31
>
> We appreciate the thoughtful review. We are pleased that our method was found to be 'well-motivated' and 'practical', and that both the effectiveness of our design and the clarity of our presentation were appreciated. We provide below a point-by-point response to each concern.
>
>
> **1. Expanded evaluation**
>
> To further strengthen the evaluation, we have added 32 additional meshes from the commercial Hyper3D website, which features AI-generated PBR textures with more diverse content. Each sample includes PBR maps at 512×512 resolution as input, and 2048×2048 resolution as GT. We perform 4× SR as in the main paper. All experimental settings remain consistent with those used on the original 16 meshes.
>
> Below we provide preliminary results comparing our method against the two strongest baselines. Results from other baselines and visualizations will be added in the final version. Our method consistently outperforms both baselines across all PBR channels and in rendering quality.
>
>
> | Method           | Albedo | Roughness | Metallic | Normal | Renderings |
> |------------------|--------|-----------|----------|--------|------------------|
> | HAT              | 26.368 | 32.750    | 31.794   | 28.235 | 28.312         |
> | CAMixerSR-FT     | 31.439 | 33.195    | 27.583   | 31.054 | 30.841           |
> | PBR-SR (ours)    | 33.841 | 35.059    | 35.352   | 34.564 | 31.208           |
>
> Importantly, the trends observed on the newly added AI-generated meshes are consistent with those on the original 16 high-quality test cases. Across both evaluation sets, our method achieves robust improvements in all PBR texture channels and final rendered appearance, further confirming its generalization ability across diverse mesh sources and content types.
>
> We will include the complete benchmark results and extended comparisons in the final manuscript.
>
>
> **2. Computational analysis**
>
> We provide a detailed discussion in Appendix Section A.2, where we report runtime across different resolutions:
> - On a single NVIDIA A6000 GPU, optimization takes approximately **30 minutes** for 2K-to-8K resolution.
> - For 1K-to-2K resolution, the process takes **less than 8 minutes** on average.
> As our method is the first to tackle mesh PBR texture super-resolution without requiring external PBR texture datasets, we adopted an optimization-based approach that requires iterative updates. This design allows us to flexibly enhance high-resolution materials for arbitrary 3D assets without retraining or domain-specific data.
>
> Our method is intended for offline SR of mesh PBR textures, and the optimized outputs can be directly loaded for real-time rendering in downstream applications. While the current optimization time is acceptable for offline use, we agree that improving efficiency is a valuable direction for future work.
>
> **3. Robustness analysis on degraded LR textures**
>
> To assess the robustness of our optimization-based method under degraded inputs, we conducted additional experiments with two common types of degradation:
> - Gaussian noise added to the LR albedo map, with σ = 5, 10, 15, 20 (in 0–255 range);
> - JPEG compression artifacts to the LR albedo map, with quality factors = 80, 60, 40, 20, 10 (lower values indicate stronger compression).
> We report both the final PSNR after optimization and the PSNR improvement from initialization (i.e., before vs. after optimization), for both the Albedo map and the rendered appearance (average over views)  on the *TableClock* mesh.
>
> | Degradation     | Albedo PSNR | Renderings PSNR | Albedo ΔPSNR from Optimization | Renderings ΔPSNR from Optimization|
> |-----------------|-------------|------------------|---------------|-------------------|
> | None            | 33.904      | 24.429           | +8.041        | +1.031            |
> | Gaussian 5      | 32.608      | 24.223           | +7.080        | +0.989            |
> | Gaussian 10     | 30.199      | 24.085           | +4.332        | +0.748            |
> | Gaussian 15     | 27.913      | 23.826           | +2.062        | +0.517            |
> | Gaussian 20     | 26.174      | 23.560           | +0.325        | +0.322            |
> | JPEG 80         | 33.554      | 24.282           | +6.873        | +0.719            |
> | JPEG 60         | 33.292      | 24.352           | +6.325        | +0.677            |
> | JPEG 40         | 32.979      | 24.323           | +6.092        | +0.648            |
> | JPEG 20         | 32.138      | 23.853           | +6.278        | +0.828            |
> | JPEG 10         | 30.974      | 22.978           | +6.006        | +0.887            |
> - Our method consistently improves both Albedo and rendering quality, even under moderate or even strong degradation.
> - Performance gradually decreases as degradation increases, which is expected, but the optimization process still provides meaningful improvements over the initialization.
> - Notably, our method remains relatively robust to JPEG compression, showing stable performance even at high compression levels.
> These results confirm that while input quality does influence the final outcome, our method can tolerate a reasonable range of degradation and still benefit from the optimization process. We will include these findings in the supplementary material.
>
> **4. Dependency on pretrained SR models and material diversity**
>
> The core objective of our work is to investigate how to robustly enhance PBR texture maps by leveraging natural image priors, without requiring large-scale PBR texture datasets.
>
> To mitigate domain gap issues, we introduce a rendering SR module that guides the optimization through rendered supervision and uses LR PBR maps as structural constraints across channels. This design allows our method to adapt and refine outputs for diverse material types without retraining.
>
> Our test set includes a wide variety of material types (e.g., metals, plastics, woods, fabrics), and results in **Table 1** show consistent improvements across all PBR channels, demonstrating the effectiveness of our strategy.
> Additionally, we conducted an ablation study using multiple pretrained 2D image SR models (DiffBIR, StableSR, and CAMixerSR), as shown in Appendix Section B: "Ablation on Image SR Models in PBR-SR" and Table 4. While DiffBIR yields the best overall performance, all models contribute to noticeable improvements in PBR texture quality, confirming the robustness of our pipeline regardless of SR model choice.
>
> **5. Discussion on failure cases/modes**
>
> Our model is generally able to generate HR PBR textures based on the provided LR PBR inputs. The main failure cases we observed occur when the input LR PBR maps are of extremely poor quality, lacking sufficient structural or textural clues. In such scenarios, the model struggles to infer plausible high-frequency details, which leads to suboptimal outputs. We have included representative failure case examples in the supplementary material to provide a clearer view of the method’s current limitations.
>
> We have further clarified this point in the revised manuscript to emphasize generalizability across material types.

---

> > ### Author Response · Authors · 2025-08-05
> >
> > Thank you again for your constructive and thoughtful review. If there is anything we can further clarify or elaborate on, we’d be happy to provide more details.

---

> > ### Comment · Reviewer_45eH · 2025-08-07
> >
> > Thank you to the authors for their efforts in addressing my concerns. The clarifications have significantly improved my understanding of the work, particularly regarding the robustness analysis on degraded LR textures and failure modes. Based on these clarifications, I am raising my score and strongly recommend that the authors incorporate these details into the main paper, which would greatly benefit readers' comprehension of the technical contributions.

---

> > > ### Author Response · Authors · 2025-08-07
> > >
> > > Thank you for the positive feedback. We will include the clarifications in the main paper.

---

### Official Review · Reviewer_rUDa · 2025-07-03

**Clarity:** 3
**Significance:** 2
**Originality:** 2
**Rating:** 5
**Confidence:** 3

**Summary:**

This paper addresses the problem of physically based rendering (PBR) super-resolution (SR) using a zero-shot, optimization-based approach that leverages a pretrained image super-resolution model. The authors use the priors from the image-based SR model to optimize the texture maps in atlas view. Supervision is provided with pseudo-labels obtained from super-resolution renders of the textured object. The loss is effectively computed between those SR renders and renders that are using the upscaled (optimized) PBR maps. Additionally, the authors adopt a weighting scheme with learnable weights (with regularizing term), a low-resolution per-map loss constraint, and a smoothness constraint on the individual loss maps. The results demonstrate the method’s capability to produce sharper and cleaner outputs.

**Questions:**

- What kind of approach do you use for rendering the images I^HR and I^SR? Are you using point lights or environment maps? Do you randomly sample these? This seems important to mention, as it will affect the optimization process.

- A naive baseline would be to take a pretrained SR model and fine-tune it for the PBR SR task. Specifically, the model would be extended to receive all maps (or maps one by one, in which case there is less to do), and output all maps at HR (or separately). Importantly, you would include a rendering loss. One could argue that such an approach would not suffer from a per-instance optimization process (cf. limitations). Wouldn't the authors agree that this is a relatively straightforward way to avoid the optimization requirement?

- Is bicubic interpolation really the correct sampling method for components such as normal and metalness maps? Metalness maps are generally binary, and normal maps represent unit vector directions.

- Fix the reference at line 69 of the supplementary material: "As shown in Fig. X". In Table 3 of the supplementary material, do the results for "w/o robust" correspond to the pixel-wise loss applied with the W weight? In this scenario, are the other loss terms (LR PBR loss and smoothness loss) still used?

**Ethical Concerns:**

["NO or VERY MINOR ethics concerns only"]

**Final Justification:**

The paper proposes a new type of super-resolution task specifically tailored for PBR map upsampling, and the presented method yields convincing results, especially in the quality and sharpness of the upsampled maps. The rendered results show that the method outperforms other approaches and is capable of tackling this new task. The authors answered all my questions raised during the rebuttal phase and were more than open to making every possible change to the overall formatting and notation of the paper. I appreciate the authors for their work and patience. After consideration, I have decided to raise my score to "accept".

**Limitations:**

I appreciate that the authors described some of the limitations and proposed possible directions for improvement. However, it would have been helpful if they had also included examples of failure cases, or at least referred to some already illustrated in the paper. Showing such examples would give a complete picture of the method's flaws.

**Paper Formatting Concerns:**

- The bibliography requires revision to ensure consistent citation formatting throughout.

- This is not specifically about formatting, but the use of voice-over breaks anonymity. I would *strongly* advise against including it in the future, as it compromises the integrity of the review process.

**Quality:**

3

**Strengths And Weaknesses:**

## Strengths

- The authors propose a new form of super resolution task, specifically tailored for PBR map upsampling.

- The method yields convincing results in PBR super resolution, especially in the quality and sharpness of the upsampled maps. The rendered views show that the method performs better than the other baselines adapted to PBR super resolution.


## Weaknesses

- The paper should be clearer about what is contained in the supplementary material (e.g., types of additional experiments and ablations, experimental specifications, and computational requirements).

- I am not very keen on the notation (k, K). As far as I know, these notations are not commonly used in the literature, though I might be mistaken. Additionally, some notations are rather peculiar, for instance, specular reflectance is usually denoted as F_0 (Cook and Torrance), and using L for the diffuse and specular terms is confusing, as L is already being used for the illumination term itself.

- I am not sure I understand the point of detailing the expression of the BRDF in the main paper. The definition of specular reflection is provided, but there is no clear link made to the BRDF model itself (Eq. 2). For readers unfamiliar with microfacet models, it may not be obvious that the specular reflection term is being used in the Fresnel term.

- The notation for the weighting regularizer (\mathcal{R}) is not convenient since it is too close to the rendering function (\mathcal{R}^M). Notations could be significantly improved to enhance the paper's readability.

- Regarding Figure 2: I find the figure a bit cluttered and think it could be clearer. While I had no issues understanding the method from the text, the figure could be more informative. Is AO omitted in Figure 2? Including the notations used in the paper, such as LR, HR, SR, and I^HR, I^SR, would also help.

- Please correct Eq. 6: there is an inconsistency between Tθ (in the equation), T_θ (line 191), and T (in the summation subscript). Also, there is an issue in the description: "the corresponding L1 texture input" is unclear.

- Listing "extensive experiments" as a standalone contribution is not meaningful, especially since it can be discussed in the introduction. While the authors are not truly introducing a new benchmark, having two clear and focused contributions is perfectly acceptable. This avoids the need for a 3rd point that does not add value as a "contribution".

- In Table 1, you identify methods relying on an optimization process with an asterisk, your method does as well. Perhaps highlighting the average runtime or optimization time would be beneficial for a fairer comparison.

---

> ### Author Rebuttal · Authors · 2025-07-31
>
> We greatly appreciate the detailed and valuable feedback from the reviewer, and we are glad that our method was found to be effective for PBR super resolution, yielding 'convincing results' with 'better quality and sharpness' than other baselines. We respond to each of the points raised below.
>
> ### Weaknesses
> **1. Supplementary material clarity**
> We will clarify the supplementary material by revising the main text to explicitly outline the contents, including detailed experimental setups, computational requirements, and additional qualitative and quantitative results.
>
> **2. Notation (k, K), F_0, and L**
> We have improved the clarity by adopting standard notations:
> - The specular reflectance is now denoted as $F_0$.
> - The symbols previously used for diffuse and specular components ($L$) have been changed to $D$ and $S$, respectively, to avoid confusion with $L$, which continues to represent illumination.
> - The notation $(k, K)$is now defined contextually or replaced with more intuitive expressions.
>
> These changes have been reflected throughout the manuscript, including the equations and figure annotations. We believe these revisions will help make the paper more accessible to readers familiar with standard PBR terminology.
>
> **3. BRDF definition clarification**
> We have now explicitly clarified the relationship between the specular reflection term and the Fresnel term to better support readers unfamiliar with microfacet-based BRDFs.
>
> **4. Notation of regularizer**
> To avoid confusion, the weighting regularizer is now explicitly denoted as $W$, clearly separating it from the rendering function $R^M$. This improves readability and interpretability.
>
> **5. Figure 2 clarity and completeness**
> We revised Figure 2 by simplifying the layout, explicitly including Ambient Occlusion (AO), and clearly labeling key components such as LR, HR, SR, $I^{HR}$, and $I^{SR}$.
>
> **6. Equation 6 consistency**
> We have corrected Equation 6 for consistent notation throughout. The ambiguous phrase “corresponding L1 texture input” has also been revised to clearly state “the corresponding L1 loss applied between textures.”
>
> **7. Contribution of extensive experiments**
> We have restructured the contribution section to focus on two core contributions. The extensive experiments are now described in the introduction to provide context, rather than being listed as a standalone contribution.
>
> **8. Table 1 optimization runtime**
> We now have included average runtime for our optimization method, and have clearly annotated runtimes for other optimization-based baselines in Table 1.
> - Paint-it SR optimization time: around 30 minutes with 2K-to-8K resolution and less than 8 minutes with 1K-to-2K. Similar optimization time as PBR-SR (Ours).
> - CAMixerSR-FT fine-tuning time: 9.5 hours of training.
> ---
> ### Questions
> **1. Rendering approach**
> We use Nvdiffrast as our differentiable renderer (line 217), and a single environment map for illumination (line 220) during the default optimization process. The environment lighting is not randomly sampled, and our default setting uses one fixed HDR environment map. We will release the code and related data.
>
> To assess robustness, we have tested with over five different environment maps. Each environment map is used individually, and the results remain consistent, as long as extreme lighting (e.g., pitch-black or overly bright) is avoided.
>
> We have further tested the training with 2 or 3 different environment maps per sample (i.e., multiple renderings per step). Even with the increased diversity, the performance is still stable.
>
> These results suggest that our method is robust to moderate lighting variation, thanks to the strong guidance from the LR PBR maps. This has been clarified in the revised manuscript.
>
> **2. Naive baseline using pretrained SR fine-tuning**
> In our paper, we compare three types of pipelines:
> - (1) *Direct inference* using pretrained 2D SR models on individual maps.
> - (2) *Fine-tuning* a 2D SR model on low-resolution PBR texture data.
> - (3) *Our optimization*, which does not rely on any external PBR dataset and optimizes only on the given textured mesh.
>
> Our method is, to the best of our knowledge, the first to address this problem setting, without any task-specific training, relying only on 2D natural image SR priors. This allows us to support arbitrary shapes and materials without requiring curated PBR datasets.
>
> We agree that your proposed approach, pretraining an SR model on texture data and using it to initialize our optimization per channel, could be a promising extension that better leverages available data. It offers a good trade-off between generalization and instance-specific refinement. We see this as a valuable direction for future work. We have clarified this discussion in the revised manuscript.
>
> **3. Interpolation method for normal and metalness maps**
>
> We appreciate the reviewer raising this point. We conducted experiments to compare different interpolation methods. Below are the results on the *Globe* mesh:
>
> | Metalness and Roughness Init | Normal Init           | Albedo | Roughness | Metallic | Normal | Renderings |
> |----------|------------------------|--------|-----------|----------|--------|------------------|
> | bicubic  | bicubic                | 35.638 | 32.002    | 35.343   | 34.299 | 36.240           |
> | bicubic  | bicubic + normalize    | 35.638 | 32.033    | 35.341   | 32.855 | 36.243           |
> | nearest  | bicubic                | 35.647 | 31.269    | 32.756   | 34.301 | 35.883           |
>
> - **Metalness and Roughness maps**: Bicubic performs slightly better than nearest-neighbor, mainly because of smoother transitions that help optimization, even for binary-valued maps.
> - **Normal maps**: Bicubic interpolation without normalization provides better performance. Interestingly, direct bicubic interpolation yields slightly better performance (i.e., PSNR 34.299 vs. 32.855). Upon investigation, we found that publicly available normal maps often have pixel vectors with norms slightly less than 1 due to compression and quantization artifacts. Normalizing after interpolation may amplify artifacts and degrade quality.
>
> Overall, bicubic interpolation serves as a robust and effective initialization, and any minor issues are corrected through our optimization process. These findings have been discussed in the manuscript.
>
> **4. Supplementary material corrections**
> - The incorrect reference at line 69 ("Fig. X") has been fixed.
> - In Table 3, "w/o robust" refers to using pixel-wise loss without the robust weighting term. Other losses are still applied. This has been clarified in the Appendix (lines 63-75).
>
> ### Limitations
> We have added failure case examples in the supplementary material, including cases with poor LR texture inputs, to give a full picture of the method’s limitations.
>
> ### Formatting and Ethical Concerns
>
> We have revised the bibliography for consistent formatting and will remove potentially identifying elements such as voice-over mentions in the future.
>
> We have clarified the above points in the revised manuscript.

---

> > ### Author Response · Authors · 2025-08-05
> >
> > Thank you again for your constructive and thoughtful review. If there is anything we can further clarify or elaborate on, we’d be happy to provide more details.

---

> > > ### Author Response · Authors · 2025-08-07
> > >
> > > Thanks again for your valuable feedback. We’ll incorporate the suggested changes and are happy to clarify anything further if needed.

---

> > > > ### Comment · Reviewer_rUDa · 2025-08-09
> > > >
> > > > A number of my comments concerned clarification, notation changes, and consistency issues, and the authors took great care in addressing these (especially the rendering equation notations). I strongly believe these changes will overall improve the readability of the paper. Regarding my two main questions, the potential naive baseline and the comparison to different interpolation modes, I believe the authors were sufficiently clear on those. It is interesting indeed that normalization after interpolation degrades performance for normals. Please do include the description of the rendering approach in your final version.
> > > >
> > > > In my opinion, the response is satisfactory and warrants acceptance.
> > > > Thank you to the authors for their effort in both the submission and the rebuttal.

---

> ### Comment · Area_Chair_NvxR · 2025-08-09
>
> Dear Reviewer,
>
> Please respond to the rebuttal from the authors. Please note that, this year it is not allowed to finalize without discussion with authors.
>
> Best,
>
> Your AC

---

### Comment · Area_Chair_NvxR · 2025-08-04
**Please discuss with the authors**

Dear Reviewers,

Please discuss with the authors, especially if the rebuttal did not solve your concerns.

Best,
Your AC

---

### Note · Authors · 2025-08-15

**Recognized Strengths by Reviewers**
We are grateful to the reviewers for their thoughtful feedback and for recognizing the novelty, practicality, and impact of this work. Highlights from the reviews include:
- **Novel problem definition:** The first formulation of PBR texture super-resolution aimed at realistic asset workflows (rUDa, MDkT, h7Gq).
- **Strong methodology:** A well-integrated combination of pre-trained diffusion priors, differentiable rendering with iterative optimization, PBR consistency constraints, TV regularization, and learnable pixel-wise loss weighting, making the approach both effective and practical (45eH, h7Gq).
- **Cross-domain distillation:** A transfer from the natural image domain to the UV domain that enables zero-shot, training-free capability, overcoming the lack of LR–HR PBR texture pairs (MDkT).
- **Empirical quality:** Higher fidelity, sharpness, and better preservation of material properties under relighting, with strong results against multiple baselines and clear ablation evidence (rUDa, 45eH, h7Gq).
- **Clarity:** Clear writing and structure that make the work easy to follow (45eH, MDkT).

---

**Review Feedback and Our Responses**
All reviewers indicated that their main concerns were resolved in the rebuttal, which led to higher scores (rUDa: “satisfactory and warrants acceptance”; 45eH: “raising my score”; MDkT: “borderline accept”; h7Gq: “revise my score from 4 to 5”).

We have already addressed, or will address in the final version, all points raised: clarifying notation and equations, improving consistency, expanding robustness and failure mode analyses, adding detailed baseline comparisons, and strengthening the ablation section.

We appreciate the constructive engagement from both reviewers and ACs, and are confident the final version will fully reflect their suggestions while retaining the strengths they highlighted.

---

### Decision · Program_Chairs · 2025-09-17

**Decision:**

Accept (poster)

**Comment:**

This submission presents a zero-shot framework for super-resolving PBR textures on 3D meshes by leveraging pretrained 2D image priors and iterative optimization with differentiable rendering. The reviewers unanimously recognized the paper's strengths, including its novel problem formulation, robust methodology, and high-quality empirical results. Key concerns raised during the review such as clarity of notation, evaluation scale, computational analysis, and robustness to input degradation, were thoroughly addressed by the authors in their rebuttal, with revisions and additional experiments (e.g., expanded datasets, failure case analysis, and runtime benchmarks) significantly improving the manuscript. The reviewers subsequently upgraded their ratings, reflecting confidence in the paper's technical soundness and impact. The submission is suggested for acceptance.